# A minimal gene set characterizes TIL specific for diverse tumor antigens across different cancer types

Zhen Zeng[1,2,3], Tianbei Zhang [1,2,3], Jiajia Zhang[4], Shuai Li[5], Sydney Connor[1,2,3], Boyang Zhang[5], Yimin Zhao[5,9], Jordan Wilson[5], Dipika Singh[1,2,3], Rima Kulikauskas [6,7], Candice D. Church[6,7], Thomas H. Pulliam[6,7], Saumya Jani[6,7], Paul Nghiem [6,7], Suzanne L. Topalian [1,3,8], Patrick M. Forde [1,3], Drew M. Pardoll [1,2,3], Hongkai Ji[5,10] & Kellie N. Smith [1,2,3,10] ✉

Identifying tumor-specific T cell clones that mediate immunotherapy responses remains challenging. Mutation-associated neoantigen (MANA) -specific CD8+ tumor-infiltrating lymphocytes (TIL) have been shown to express high levels of *CXCL13* and CD39 (*ENTPD1*), and low IL-7 receptor (*IL7R*) levels in many cancer types, but their collective relevance to T cell functionality has not been established. Here we present an integrative tool to identify MANA-specific TIL using weighted expression levels of these three genes in lung cancer and melanoma single-cell RNAseq datasets. Our three-gene "MANAscore" algorithm outperforms other RNAseq-based algorithms in identifying validated neoantigen-specific CD8+ clones, and accurately identifies TILs that recognize other classes of tumor antigens, including cancer testis antigens, endogenous retroviruses and viral oncogenes. Most of these TIL are characterized by a tissue resident memory gene expression program. Putative tumor-reactive cells (pTRC) identified via MANAscore in anti-PD-1-treated lung tumors had higher expression of checkpoint and cytotoxicity-related genes relative to putative non-tumor-reactive cells. pTRC in pathologically responding tumors showed distinguished gene expression patterns and trajectories. Collectively, we show that MANAscore is a robust tool that can greatly enrich candidate tumor-specific T cells and be used to understand the functional programming of tumor-reactive TIL.

Tumor-infiltrating lymphocytes (TIL) are integral to the tumor microenvironment (TME) and exert a central influence on immunotherapy response. Therapeutic activation of T cells targeting mutation-associated neoantigens (MANA) can be achieved through blockade of T cell checkpoints, such as PD-1[1]. Also, T cells specific for viral oncogene products in virus-induced cancers as well as self-antigens upregulated in tumors relative to normal cells—so called tumor-associated antigens (TAA)—have been identified, which can further contribute to PD-1 blockade-induced anti-tumor responses. It is well recognized that the state of function or dysfunction of tumor-specific

[1]Bloomberg~Kimmel Institute for Cancer Immunotherapy, Baltimore, MD, US. [2]Mark Center for Advanced Genomics and Imaging, Baltimore, MD, US. [3]Department of Oncology, Johns Hopkins University School of Medicine, Baltimore, MD, US. [4]David Geffen School of Medicine, University of California, Los Angeles, CA, US. [5]Department of Biostatistics, Johns Hopkins Bloomberg School of Public Health, Baltimore, MD, US. [6]Fred Hutchinson Cancer Center, Seattle, WA, US. [7]Department of Medicine, University of Washington, Seattle, WA, US. [8]Department of Surgery, Johns Hopkins University School of Medicine, Baltimore, MD, US. [9]Present address: Department of Biostatistics, University of Washington, Seattle, WA, US. [10]These authors contributed equally: Hongkai Ji, Kellie N. Smith. ✉e-mail: kellie@jhmi.edu

TIL in the TME, as defined by gene expression programs, is a major factor in anti-tumor immunity and defining genetic programs associated with immunotherapeutic response is a key endeavor in cancer immunology. A major challenge in this effort is the definition of gene expression patterns among true tumor-specific T cells, which usually represent a tiny proportion of T cells within tumors.

Recent efforts have focused on directly quantitating the frequency and breadth of the tumor-specific T cell repertoire, and transcriptional programs determining the functional state of true tumor-specific tumor infiltrating lymphocytes (TIL). By performing whole exome sequencing, neoantigen predictions, and testing 1200 tetramers on 40 untreated NSCLC TIL samples, Simoni and colleagues identified a single tetramer+ population of MANA-specific TIL from each of two patients and found that they highly expressed CD39[2], a finding that has been consistently confirmed in follow up studies[3–7]. However, the limited number of unique MANA-reactive TIL identified in this study precluded the definition of broader consistent transcriptomic signatures associated with tumor recognition. This and other studies demonstrated not only the technical challenges of tumor-specific T cell identification, but also that they represent a small proportion of total TIL, most of which are "bystanders" percolating through the tumor but without specificity for the tumor cells themselves. We and Oliveira et al. reported that integrated single-cell RNAseq and TCRseq (scTCRseq/RNAseq) combined with assays to validate tumor specificity can be used to characterize tumor-specific TIL in NSCLC and melanoma[8,9]. Several studies have documented transcriptomic features that are distinct in tumor-specific T cells among global TIL, including but not limited to higher levels of *CXCL13*[10] and *ENTPD1* (CD39)[2,11] and genes encoding multiple immune checkpoints associated with both T cell activation and "exhaustion" or "dysfunction", as well as lower expression of the IL-7 receptor (*IL7R*).

Importantly, our previous work in lung cancer showed that MANA-specific TIL can be readily detected, even from non-responding tumors, thus underscoring the importance of studying the transcriptional and functional programming of these cells as a function of therapeutic response. Our initial findings, as well as those of others[6,9,12], suggested that a comprehensive transcriptional profile of MANA-specific TIL could be used to identify or at least "enrich" for bona fide tumor-reactive T cells. Others have proposed that as few as a single gene, *CXCL13*[10], or as many as 243 genes[12], could prospectively identify tumor-reactive CD8+ TIL; an analogous gene set can also identify tumor-reactive T cells in peripheral blood[13].

Herein, we began by integrating the coupled scTCRseq/RNAseq Caushi et al. lung cancer and Oliveira et al. melanoma datasets[8,9], each with functionally-validated MANA-specific TIL, to develop and validate 'MANAscore', a bioinformatic scoring algorithm based on weighted expression of only 3 genes whose expression has previously been shown to mark tumor-reactive TIL: *CXCL13*, *ENTPD1*, and *IL7R*. The MANAscore exhibited superior performance than the aforementioned published signatures in distinguishing MANA-specific TIL from bystander TIL. In addition to identifying neoantigen-specific TIL, application of MANAscore to publicly available datasets[12,14] showed that it also identifies TAA-specific and human endogenous retrovirus (HERV)-specific TIL. Finally, application of MANAscore to a single-cell transcriptomics dataset of TIL from Merkel cell polyomavirus (MCPyV)-induced Merkel cell carcinoma (MCC) identified tetramer-validated MCPyV-specific T cells with significantly high sensitivity, thus supporting the use of MANAscore to identify TIL that recognize several different classes of tumor antigens, not just mutation associated neoantigens. By then applying MANAscore to the anti-PD-1 treated resectable lung cancer dataset, we found the majority of putative tumor reactive cells (pTRC) exhibited a tissue resident memory (TRM) transcriptional signature and that the MANAscore transcriptional program was driven by the tumor microenvironment. pTRC from pathologically responding tumors were characterized by higher

expression of genes associated with a progenitor exhausted/stem-like phenotype such as *CCR7* and *IL7R*, whereas pTRC from non-responding tumors were enriched in expression of genes associated with transitory effector T cells that precede terminal differentiation, including *LAG3*, *TIGIT*, *IFNG*, and *GZMB*.

Collectively, our data show that a 3-gene transcriptional signature can identify CD8+ TIL that recognize epitopes derived from diverse classes of tumor antigens, and that this MANAscore can be used to discover new biology related to checkpoint blockade response. The parsimonious nature of our 3-gene score opens the possibility that MANAscore could be implemented spatially for scientific discovery or for assessment as a predictive biomarker.

## Results
### Transcriptional programs of MANA-specific TIL are consistent among melanomas and lung cancers

To investigate the transcriptomic landscape of tumor-specific TIL across two tumor types with distinct clinical and biological characteristics, we performed an integrated transcriptomic analysis of CD8+ TIL at the single cell level from resectable NSCLCs and melanomas from which large numbers of MANA-specific TIL were previously identified and validated (Supplementary Data 1,2)[8,9]. Tumor specimens from 18 patients (15 NSCLC and 3 melanoma) were previously sequenced using coupled scTCR and scRNA sequencing (scTCRseq/RNAseq). These two datasets contain the largest number of functionally validated fresh (i.e., not cultured ex vivo) tumor-reactive CD8+ TIL thus far published. All NSCLC patients received neoadjuvant anti-PD-1 (nivolumab, NCT02259621) and melanoma patients received personalized neoantigen vaccines (NCT01970358). These datasets provide an opportunity to compare and contrast the transcriptional signatures that define tumor-reactive TIL across tumor types. For this integrated analysis, single cell transcriptomes were analyzed through an identical batch corrected, imputation, filtering, and cluster annotation pipeline.

A total of 253,161 CD8+ TIL were included, of which 222,987 were from the NSCLC cohort and 30,174 were from the melanoma cohort (Fig. 1A, B; Supplementary Fig. 1A). Fourteen unique T cell clusters were identified using unsupervised clustering, including six tissue resident memory (TRM) clusters (based on high expression of *ITGAE* (CD103) and *ZNF683* (HOBIT)), two effector clusters, a proliferating cluster, a CD4/CD8 double-positive cluster, a naïve cluster, and a mucosal-associated invariant T cell (MAIT) cluster, and were visualized by UMAP. Expression of subset-defining markers and T cell checkpoints was visualized for each cluster (Fig. 1C; Supplementary Fig. 1B). CD8+ TIL from melanoma were more enriched in the TRM (IV) cluster, which is characterized by upregulated expression of *CXCL13*, *CRTAM*, *TNFRSF9* (4-1BB), *XCL1/2* and *FABP5*, whereas those from NSCLC had greater representation of TRM (II) with markers of both cytotoxicity and dysfunction/exhaustion (co-expression of *GZMB*, *GZMH*, *PDCD1*, and *CTLA4*) (Supplementary Fig. 1A; Supplementary Data 3, 4).

MANA-specific T cells as identified previously (Supplementary Data 2)[8,9] were detected at variable frequencies among CD8+ TIL from nine patients (median = 1.18%, range 0.01%–35.8%). The localization of these cells onto the integrated global CD8+ TIL UMAP was strikingly similar across melanoma and NSCLC (Fig. 1D). Importantly, the majority of MANA-specific TIL resided in similar TRM clusters in both cancers (84.62% for NSCLC and 59.78% for melanoma; Fig. 1D, E), characterized by expression of *CD103*, *CXCL13*, *ZNF683*, and *ENTPD1* (CD39). Oliveira et al. annotated the majority of the MANA-specific TIL as exhausted based on high expression of genes encoding immune checkpoints that were previously demonstrated to be up-regulated in exhausted CD8+ T cells in the setting of chronic viral infection[15]. A much larger proportion of MANA-specific CD8+ TIL resided in effector clusters in melanoma than in NSCLC TIL (Fig. 1E; Supplementary Data 5; *p* = 0.025). In melanoma, CD8+ TIL specific for non-mutated melanoma-associated antigens (MAA) resided in a distinct cluster

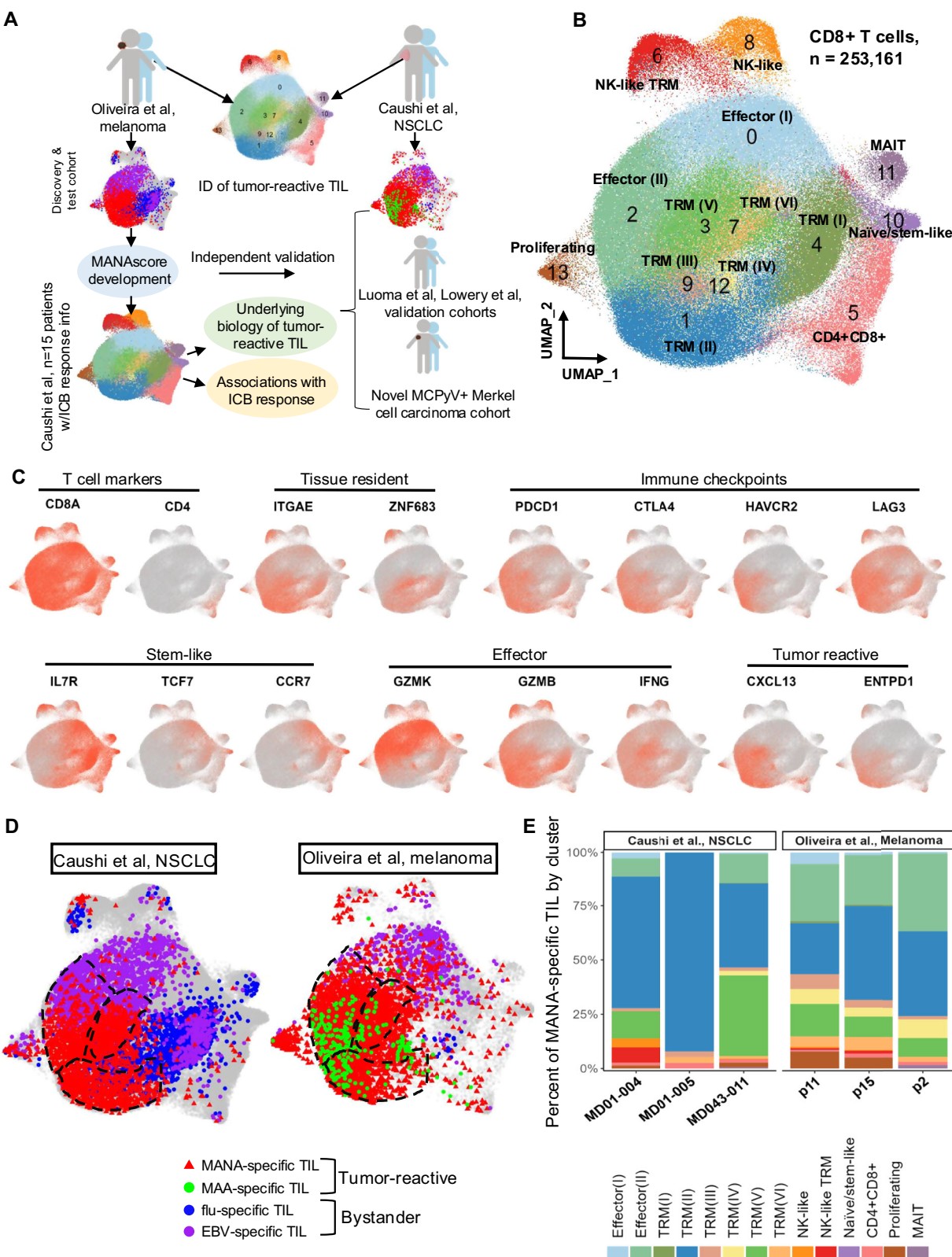

**Fig. 1 | Integration of published NSCLC and melanoma datasets. A** Study schema. **B** UMAP projection of the expression profiles of 253,161 CD8+ T cells from melanoma (*n* = 3 tumors) and NSCLC (n = 15 tumors, 12 normal lung). Immune cell subsets, defined by 14 unique clusters, are annotated and marked by color code. **C** Expression of selected genes including T cell marker genes (*CD8A*, *CD4*), tissue resident genes (*ITGAE*, *ZNF683*), stem related genes (*IL7R*, *TCF7*, *CCR7* and *SELL*), tumor reactive genes (*CXCL13* and *ENTPD1*), effector related genes (*GZMK*, *GZMB*, *PRF1*, *GNLY* and *IFNG*) and immune checkpoint genes (*PDCD1*, *CTLA4*, *TIGIT*,

*HAVCR2* and *LAG3*). **D** MANA-specific (red), MAA-specific (green), influenza-specific (blue) and EBV-specific (purple) clonotypes as identified in the cited publications were overlaid on the CD8+ UMAP. **E** Proportion of MANA-specific T cells in tumor tissues (tumor bed for melanoma) by cell type for six patients (3 NSCLC and 3 melanoma) with more than 30 functionally validated MANA-specific CD8+ TIL identified. Figure 1A was created in BioRender. Smith, K. (2024) https://BioRender.com/n17a163.

straddling effector and TRM regions (Fig. 1D). EBV-specific T cells were prevalent in NSCLC and melanoma TIL and mapped almost exclusively to effector T cell clusters. While melanomas contained relatively few flu-specific T cells compared with NSCLC (consistent with flu being a respiratory virus), they mapped to a unique TRM cluster distinct from the clusters occupied by MANA-specific T cells in both tumor types. Together, these data provide a global transcriptomic landscape of MANA-specific CD8+ TIL in NSCLC and melanoma and highlight notable similarities and differences in transcriptional programs between TIL from these two tumor types.

## A 3-gene signature distinguishes MANA-specific from virus-specific CD8+ TIL

Analyzes of the above-described recent single cell studies in our NSCLC cohort and melanoma cohort directly identifying and functionally validating MANA-, EBV-, and Influenza (flu)-specific T cell clones at the TCR level[8,9] provided a unique ground truth dataset from which to test and validate gene expression patterns potentially specific for MANA-specific CD8+ TIL. Given the significant transcriptomic similarities between MANA-specific TIL in NSCLC and melanoma, we hypothesized that conserved gene programs may distinguish MANA-specific from bystander TIL across tumors. Of those genes which are differentially expressed in MANA-specific TIL vs EBV-/flu-specific TIL, *CXCL13* emerged as an effective marker for identifying tumor-reactive T cells within the TME[10]. *ENTPD1* (CD39), a gene associated with T cell exhaustion, was previously nominated as a marker for tumor-reactive TIL in lung cancer[2] and indeed was upregulated in tumor-reactive TIL relative to bystander TIL in the melanoma dataset used here[2,11]. Lastly, *IL7R*, which homeostatically maintains memory T cells, was notably high in flu-specific T cells but low in melanoma MANA-specific TIL (Fig. 2A)[9].

Based on our a priori knowledge of these genes as individual markers of anti-tumor reactivity, we used them to construct single-patient models trained using 80% of the data from MANA- and EBV-/flu-/non-tumor antigen-specific TIL in three melanoma patients. SAVER imputed (MANAscore_i) and non-imputed (MANAscore_ni) combined voting models were generated for predicting MANA-specific TIL in unseen data. These two models were applied to a test dataset consisting of the remaining 20% of all MANA-reactive TIL and 20% of all EBV-/flu-/non-tumor antigen-specific TIL from the melanomas (Fig. 2B; Supplementary Fig. 2A; MANAscore_i AUC: 0.89; MANAscore_ni AUC: 0.80), and, separately, all TIL specific for MAAs and 20% of the EBV-/flu-/non-tumor antigen-specific TIL in melanoma patient p2 (Fig. 2B; Supplementary Fig. 2A; AUC: MANAscore_i: 0.87; MANAscore_ni: 0.80). The performance of both imputed and non-imputed 3-gene MANAscore models were consistently superior to the 243-gene NeoTCR8 calculated by scGSEA[12]. Overall, the imputed 3-gene MANAscore performed better than single gene models of *CXCL13*, *ENTPD1*, and *IL7R* alone. The models were further assessed on an independent validation cohort of 72 and 38 MANA-specific TIL from lung cancer patients MD01-004 and MD01-005 with NSCLC, respectively (Fig. 2C; Supplementary Fig. 2B). Both the imputed and non-imputed MANAscores (MANAscore_i AUCs: 0.96, 0.96; MANAscore_ni AUCs: 0.97, 0.94) were consistently superior to NeoTCR8 (AUCs: 0.86, 0.83) and comparable to or better than the single gene models (AUCs: 0.61–0.96). Like melanoma, MANA-specific TIL in NSCLC showed higher expression of CXCL13 and ENTPD1, and lower expression of IL7R compared with virus-specific TIL (Supplementary Fig. 2C). Application of our combined voting model to the NeoTCR8 gene signature improved its performance over the original scGSEA model as published by Lowery et al. [12] (Supplementary Fig. 2D). While this led to improved performance over MANAscore in the melanoma test data, representing the same tumor type as the training data, MANAscore still outperformed NeoTCR8 (combined voting and scGSEA models) in the independent NSCLC validation data, representing a tumor type different from the training data (Supplementary

Fig. 2D). Projecting MANAscore as a continuous scale (range from 0 to 1) onto the CD8+ TIL UMAP, we observed consistent distribution of MANAscore[hi] TIL within the TRM clusters and co-localization with functionally validated MANA-specific clones (Fig. 2D). Additionally, the MANAscore[hi] cells in the effector II cluster co-localized with validated melanoma MANA-specific TIL that resided in this cluster. Taken together, these results suggest that MANAscore could be an enhanced tool for de novo identification of tumor-specific clones across transcriptional clusters.

We next wanted to refine the positive predictive value by determining a specific threshold at which to define MANAscore[hi] cells. The MANAscores of CD8+ TIL exhibited substantial variability across patient's tumors, with the mean MANAscore_i and MANAscore_ni ranging from 0.205 in NSCLC patient NY016-015 to 0.718 inpatient MD01-010 (Supplementary Fig. 3). Even within individual patient tumors, there existed a considerable range of MANAscores, suggestive of the different functional states of TIL within the TME. These two sources of heterogeneity, both across and within individual patients, presented a challenge for selecting a threshold for identifying MANAscore[hi] CD8+ TIL. To address this heterogeneity, we defined the MANAscore[hi] TIL for each patient according to the distribution of both the imputed and non-imputed MANAscores (Fig. 2E; Supplementary Data 6), with the cutoff set at the last trough observed for both MANAscore distributions. Using both of these scores, rather than just a single imputed or non-imputed score, increases our confidence that the identified cells may be true tumor-reactive TIL. Once we identified MANAscore[hi] TIL, we observed that 84.98% were found in either the TRM(II), Effector (II), or TRM(V) clusters (Fig. 2F).

## MANAscore can be used to define putative MANA-specific clones
A fundamental feature of the T cell receptor (TCR) is that T cells with the same TCR will always recognize the same antigen[16]. The utilization of coupled scTCR-seq in the Caushi et al., scRNAseq dataset enabled us to study the intraclonal MANAscore heterogeneity, for all cells with the same TCR corresponding to a MANAscore[hi] cell. This heterogeneity was substantial (Supplementary Fig. 4A), potentially due to different activation states, antigen encounters, and microenvironment exposures within tumor-specific CD8+ clones. In total, we identified 443 clones out of a total of 18,818 clones (2.37%) of undefined antigen specificity among our NSCLC TIL dataset (Supplementary Data 7), in which at least 5 cells were MANAscore[hi] TIL. 45,190 of the 145,652 total CD8+ NSCLC TIL (37.21% of all CD8+ NSCLC TIL profiled) were termed 'putative tumor-reactive cells' pTRC because they shared a TCR with at least 5 MANAscore[hi] TIL. Of the 45,190 pTRC, 17,382 (38.46%) were MANAscore[hi] and 27,808 (61.64%) were MANAscore[lo] (Supplementary Fig. 4B). MANAscore[hi] pTRC occupied the same regions of the UMAP as functionally validated MANA-specific TIL (Fig. 3A). While including MANAscore[lo] pTRC resulted in a more diffuse distribution within the UMAP (Supplementary Fig. 4C), 79.12% of pTRC still localized to clusters TRM(II), Effector (II), and TRM(V) (Supplementary Fig. 4D). Additionally, 4.10% of pTRC localized to the Eff(I) cluster (Supplementary Fig. 4D), thereby demonstrating the heterogeneity of T cells sharing the same TCR. In addition, there was a large proportion of pTRC TCR clones was shared among the TRM(II), Effector (II) and TRM (V) clusters (Supplementary Fig. 4E).

To further illustrate the predictive robustness of our model across multiple tumor types and tumor antigen classes, we applied MANAscore to previously published single T cell transcriptomic data from TIL derived from the Luoma et al. oral cancer[14] and Lowery et al. metastatic cancers (including breast cancer, colorectal cancer, and melanoma)[12] studies. The Luoma et al. study identified tumor-associated antigens including a cancer-testes antigen and a human endogenous retroviral antigen while the Lowery et al. focused on classical MANA. The known TAA (*n* = 136) and MANA-specific (*n* = 303) TIL, respectively, were precisely situated within the MANAscore[hi]

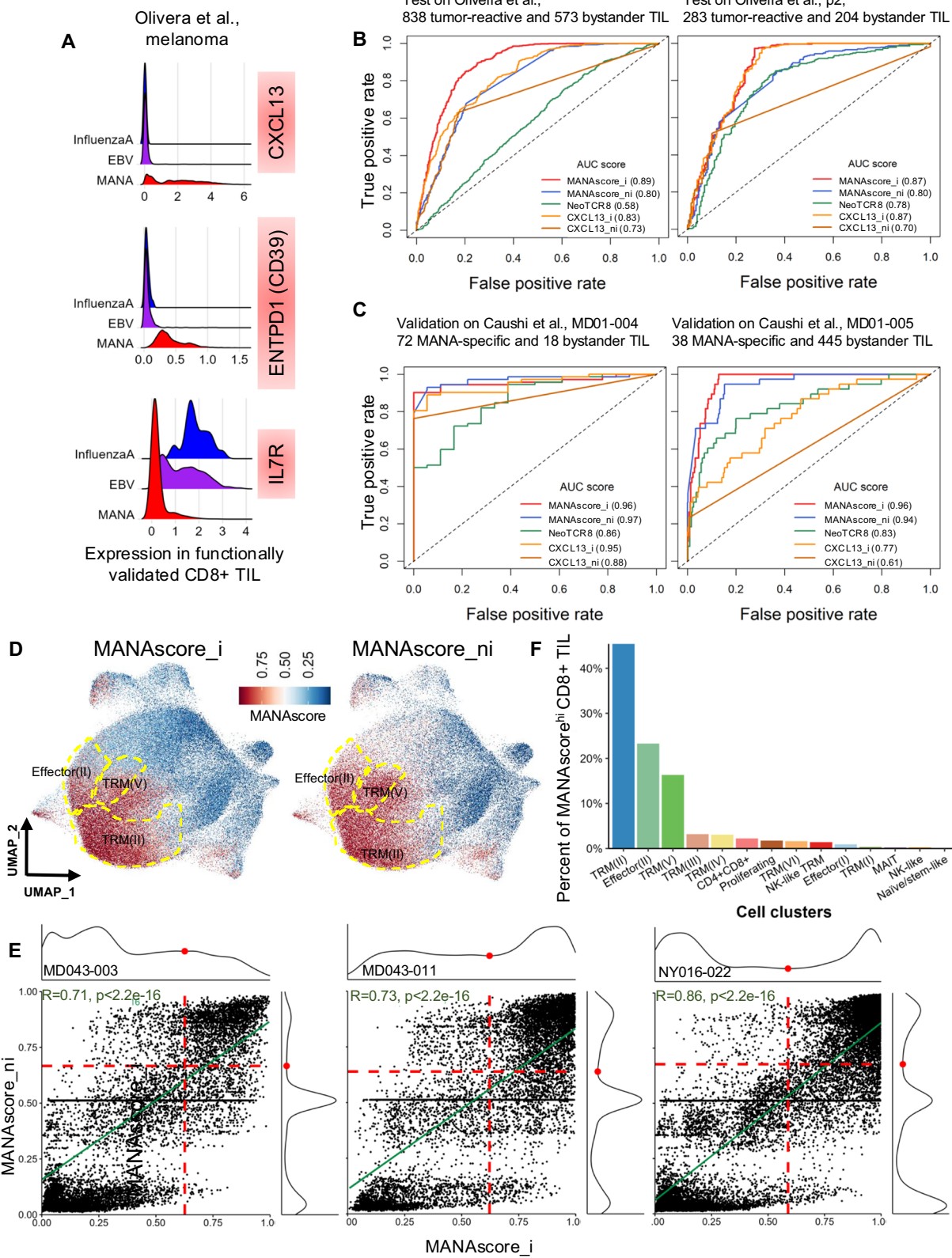

regions of the UMAP (Fig. 3B; Supplementary Fig. 5, 6). We then tested if application of our MANAscore method to identify pTRC would capture the previously-defined tumor-reactive clones from these studies. Indeed, all validated tumor-reactive clones ($n = 4$) from 3 early-stage oral cancer patients were captured by MANAscore trained by melanoma (Supplementary Fig. 5C). From the Lowery et al. metastatic cancers study, of all 31 MANA-specific clones in 9 patients with

metastatic cancers, 17 clones in 6 patients (26 MANA-specific clones in total) were captured by our method (Supplementary Fig. 6C). This lower rate capture rate may be due to the metastatic nature of these cancers relative to those studied by Luoma et al. and our cohort of resectable NSCLC. This difference may contribute to the lower performance of NeoTCR8, which was derived from the Lowery et al. data, on the melanoma and lung cancer data in our present study (Fig. 2B, C).

**Fig. 2 | Development of MANAscore using patient-specific thresholds.**
**A** Expression level of three genes previously reported to mark tumor-reactive vs bystander CD8+ TIL in validated MANA-, EBV- and InfluenzaA-specific TIL in the Oliveira et al melanoma dataset. **B** Receiver operating characteristic (ROC) curves for the performance of different models, including imputed MANAscore (MANA-score_i), non-imputed MANAscore (MANAscore_ni), original NeoTCR8 score (calculated by scGSEA), imputed CXCL13 single gene model (CXCL13_i), and non-imputed *CXCL13* single gene model (CXCL13_ni) on the melanoma test data. Test data consisted of 20% of the validated MANA- and EBV-/InfluenzaA- specific CD8+ TIL (*n* = 838 tumor-reactive and 573 bystander TIL) and all melanoma associated antigen (MAA)-specific and 20% of the EBV-/InfluenzaA-specific CD8+ TIL from melanoma patient p2 (*n* = 283 tumor-reactive and 204 bystander TIL). **C** ROC curves for the performance of the MANAscore_i, MANAscore_ni, NeoTCR8,

CXCL13_i, and CXCL13_ni models on the validation datasets. Validation was performed on 72 MANA-specific and 18 bystander CD8+ TIL from NSCLC patient MD01-004 and on 38 MANA-specific and 445 bystander CD8+ TIL from NSCLC patient MD01-005 from Caushi et al. **D** Feature plot for MANAscore_i (left) and MANAscore_ni (right) on CD8 TIL with unknown antigen specificity in the Caushi et al NSCLC cohort. MANAscore ranges from 0 (blue) to 1 (red). **E** Scatter plot of MANAscore_i and MANAscore_ni in selected NSCLC patient tumors. Cutoffs were set to define MANAscore[hi] CD8+ TIL by assessing distribution of these two scores at the patient level. Red dotted lines indicate the thresholds for MANAscore[hi] in the imputed and non-imputed models, the linear correlation of these two scores was marked in green, Pearson's correlation coefficient is added. **F** Cell type contribution to MANAscore[hi] CD8+ TIL.

We then sought to determine whether the MANAscore could predict CD8+ T cell responses against a very different class of tumor-specific antigen derived from a viral oncogene. This was of particular interest because it was unknown if the transcriptional programming of TIL specific for a tumor-associated viral oncogene could more closely resemble MANA-specific TIL or other virus-specific TIL (influenza, EBV). CD8+ TIL (*n* = 43,380) from three Merkel cell polyomavirus (MCPyV)-driven MCC were subjected to scRNA/TCR-seq and MCPyV antigen specific T cells were identified using CITEseq with various barcoded MCPyV peptide-HLA multimers matched to each patient's HLA genotype[17] (Supplementary Data 8). Eight unique T cell clusters were identified, including two TRM clusters, GZMK[+], intermediate, stem-like, NK-like, proliferating and FoxP3[+] clusters (Fig. 3C; Supplementary Fig. 7A; Supplementary Data 9). The MCPyV-specific TIL were identified through MCPyV pep-HLA CITEseq according to multimer distribution (log transformed) (Supplementary Fig. 7B), resulting in the identification of 823 MCPyV-specific TIL (Supplementary Fig. 7C). Similarly, those TIL sharing the identical TCR with the multimer-positive TIL were considered to be MCPyV-specific as well. Ultimately, 4473 MCPyV-specific TIL were identified (roughly 10% of the 43,880 total CD8+ TIL) across 33 clones (Fig. 3D), which exclusively existed in MANAscore[hi] regions of the UMAP (Fig. 3E). Then we used the same method as in Fig. 2E to define MANAscore[hi] TIL for each patient based on the distribution of both MANAscore_i and MANAscore_ni. All but 7 MCPyV-specific TIL (99.86%) were captured by our MANAscore (Fig. 3F). Using the same methods to identify pTRC, we identified 8640 pTRC from 74 clones, including 5405 MANAscore[hi] pTRC identified (Fig. 3G; Supplementary Data 10), with 99.05% of the MCPyV-specific TIL (clonal size ≥ 5) included (Fig. 3H), which strongly supports the robustness and sensitivity of MANAscore in identifying CD8+ TIL specific for tumor-associated viral oncogenes. Taken together, our analyzes validate the potential for MANAscore to pinpoint tumor-reactive CD8+ T cells among TIL from highly diverse cancers and specific for diverse categories of tumor antigen.

We next wanted to study the underlying transcriptional programs of pTRC as identified using MANAscore. T cells undergo clonal expansion upon engagement of their TCR with its cognate peptide:MHC ligand. Indeed, pTRC identified via MANAscore had a significantly larger clonal size than non-pTRCs (*p* < 2.22e-16) (Fig. 4A, B; Supplementary Fig. 8A, B), demonstrating that pTRC in the TME are antigen exposed and have clonally proliferated in response. Not surprisingly, the 3 genes used to compute MANAscore were the genes significant differentially expressed in pTRC relative to non-pTRC (Fig. 4C; Supplementary Data 11). More importantly, we also observed the upregulation of the T cell checkpoints *CTLA4* and *HAVCR2* (TIM-3), as well as *GZMB* and *GNLY* in pTRC, along with downregulation of *GZMK* and *CCR7*. This was consistent with the hybrid transcriptional program identified previously in validated neoantigen-specific TIL[8] and the phenotype of 'terminal effector' CD8+ TIL identified by Galletti et al.[18] pTRC also showed upregulation of *RBPJ*, which is involved in the *RBPJ-IRF1* axis that promotes terminal differentiation of T cells[19].

Higher TRM signature scores were noted in pTRC compared with non-pTRC (Fig. 4D), consistent with the predominant localization of MANA-specific TIL within TRM clusters[8]. Unsurprisingly, pTRC displayed higher checkpoint[8], cytotoxicity[14], and TCR signaling[20] scores relative to non-pTRC (Fig. 4E–G; Supplementary Data 12), which aligns with the characteristics of tumor-specific T cells owing to their persistent antigen exposure and activation state[21]. There was a strong correlation (R = 0.86, *p* < 2.2e-16;) between the ranks of differentially expressed genes identified in validated MANA- vs flu/EBV-specific TIL and pTRC vs non-pTRC, thus supporting the high similarity between MANA-specific TIL and pTRC in the tumor (Fig. 4H) and supporting the notion that our method can precisely recapitulate known transcriptional differences between clones recognizing tumor antigens vs influenza or EBV antigens.

## Impact of the TME on pTRC functional programming

We then wanted to assess if the 3-gene MANAscore was largely determined by factors specific to the TME. Because our selection of MANAscorehi TIL is based on both imputed and non-imputed scores, we first calculated the average overall MANAscore of T cells from the tumor and adjacent normal lung (*n* = 12) using $\sqrt{\mathrm{MANAscore\_i}^2 + \mathrm{MANAscore\_ni}^2}$. Interestingly, while there are very minor inconsistencies in which MANAscore (imputed or non-imputed) as better performance in the 3 independent validation datasets (oral cancer, metastatic cancers, and MCC), the combined overall MANAscore is consistently closer to the better performing score (Supplementary Fig. 9). The average overall MANAscore in adjacent normal lung was markedly lower compared with tumor (*p* = 0.00098; Fig. 5A; Supplementary Fig. 10A), and indeed, when applying the MANAscore to T cells derived from adjacent normal lung of patients MD01-005 and MD043-011, none of the validated tumor-reactive clones were captured (Supplementary Fig. 10B). Thus, the MANAscore is highly effective in detecting tumor-specific T cells found within tumor tissues but is largely dependent on the TME. This finding is consistent with a prior report showing differential transcriptional programming in favor of less "dysfunction" in circulating tumor-reactive cells[13]. Using the TCR as a barcode, we were able to track pTRC and non-pTRC (as identified in the tumor tissue) in adjacent normal lung and observed that pTRC were enriched in NSCLC relative to normal lung (Fig. 5B). Checkpoint, cytotoxicity, and TRM signature scores, as well as *ENTPD1* expression, were also higher in pTRC from the tumor, and stemness scores were lower in tumor pTRC (Fig. 5C–F; Supplementary Fig. 10C–D).

## Gene expression differences among pTRC associated with response to PD-1 blockade

We previously did not observe a difference in frequency of functionally validated tumor-reactive CD8+ T cells associated with pathologic response to neoadjuvant PD-1 blockade in the Caushi et al. resectable NSCLC dataset[8]. However, the approaches we used in that study to identify neoantigen-specific T cells, combining functional expansion from peripheral blood, scTCRseq and TCR transfer into Jurkat reporter

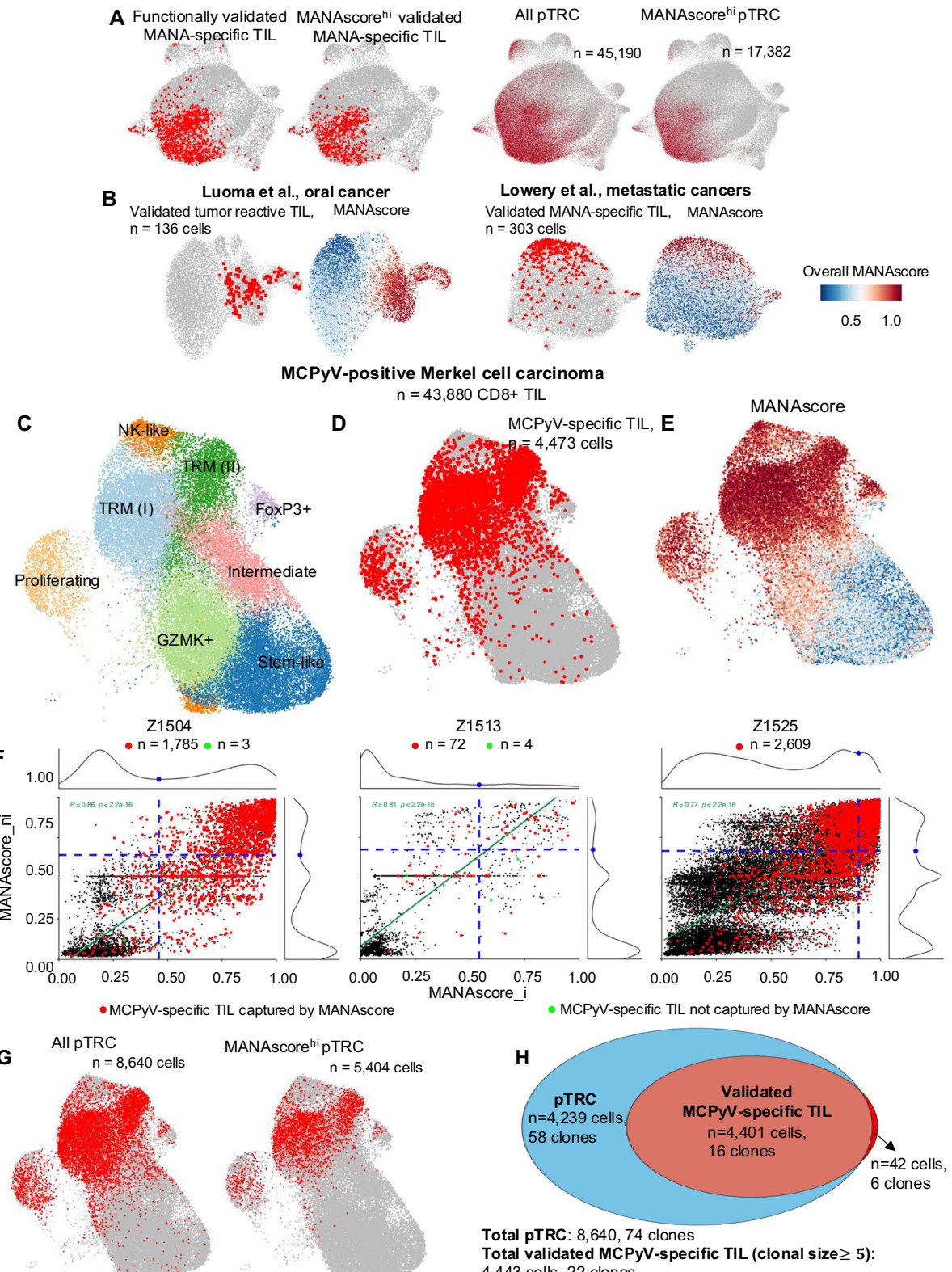

cells, may have missed a significant proportion of total tumor-reactive cells. We re-examined this issue by using MANAscore to identify pTRC, thus providing the opportunity to query a much larger number of single cell transcriptomes. We compared the frequency of MANAscore$^{hi}$ TIL and pTRC in a binary fashion with major pathologic responders (MPR, ≤10% of residual viable tumor cells after anti-PD-1 therapy) vs. non-MPRs, or as a linear function of percent residual viable

tumor in 10% increments. The proportion of MANAscore$^{hi}$ TIL in the tumor trended towards being higher in responding tumors and was weakly correlated with percent residual viable tumor ($p = 0.140$ and 0.086, respectively; Supplementary Figs. 11A, B). While there was also no difference in the proportion of MANAscore$^{hi}$ TIL from specific clusters in responding and non-responding tumors (Supplementary Fig. 11C), the pseudobulk gene expression profile of MANAscore$^{hi}$ TIL

**Fig. 3 | MANAscore can identify putative tumor-reactive TIL. A** Cell overlays to show the UMAP localization of functionally validated MANA-specific TIL, functionally validated MANAscore[hi] MANA-specific TIL, all pTRC ($n = 45{,}190$), and MANAscore[hi] pTRC ($n = 17{,}382$) on the CD8+ UMAP. **B** Functionally validated tumor-reactive TIL ($n = 136$ cells) overlaid on the CD8+ TIL UMAP of the Luoma et al. oral cancer cohort (top left) and the overall MANAscore ($\sqrt{\text{MANAscore}\_i^2 + \text{MANAscore}\_ni^2}$) predicted for CD8+ TIL in the same cohort (top right). Functionally validated MANA-specific TIL ($n = 303$ cells) overlaid on the CD8+ TIL UMAP of the Lowery et al. metastatic cancer cohort (bottom left) and the overall MANAscore ($\sqrt{\text{MANAscore}\_i^2 + \text{MANAscore}\_ni^2}$) overlaid on the same UMAP (bottom right) is also shown. MANAscore ranges from 0 (blue) to 1 (red). **C** UMAP projection of the expression profiles of 43,880 CD8+ T cells from three Merkel cell carcinoma patients with MCPyV-positive. Immune cell subsets, defined by 9 unique clusters, are annotated and marked by color code. **D** CITEseq identified MCPyV-specific TIL of high quality ($n = 4473$ cells) overlaid on the CD8+ TIL UMAP of Merkel cell carcinoma cohort from patients with MCPyV positive. **E** Overall MANAscore ($\sqrt{\text{MANAscore}\_i^2 + \text{MANAscore}\_ni^2}$) predicted for CD8+ TIL in the same cohort (right). **F** Scatter plot of MANAscore_i and MANAscore_ni for TIL in three Merkel cell carcinoma patient. The TIL captured by both CITEseq and MANAscore were marked red ($n = 4466$ in total), the TIL only captured by CITEseq were marked green ($n = 7$). The linear correlation of these two scores was marked in green, Pearson's correlation coefficient is added. **G** pTRC and MANAscore[hi] pTRC identified overlayed on the CD8+ UMAP of Merkel cell carcinoma. **H** Venn diagram of TIL identified by MANAscore (pTRC, $n = 8640$) and CITEseq (MCPyV-specific TIL, $n = 4473$), of which 4401 (99.05%) TIL were identified by both methods.

showed a marginally significant canonical correlation with percent residual tumor (canonical correlation: 0.59, $p = 0.05$) (Supplementary Fig. 11D), but not with discrete response (MPR vs non-MPR) to PD-1 blockade (canonical correlation: 0.48, $p = 0.18$) (Supplementary Fig. 11D). When looking at all pTRC, however, their proportion was higher in responding relative to non-responding tumors ($p = 0.047$; Fig. 6A) and was negatively correlated with percent residual tumor (Fig. 6B; R = −0.55, $p = 0.035$), compatible with the notion that MANA-specific cells are important drivers of anti-tumor immunity during PD-1 blockade. To next test the hypothesis that, in addition to the overall abundance of pTRC, the MANAscore itself could be used to categorize responding vs non-responding tumors after anti-PD-1 therapy, we computed the average overall MANAscore for each tumor sample and found a higher, albeit not statistically significant, average MANAscore in MPR than non-MPR tumors ($p = 0.088$; Fig. 6C). The pseudobulk gene expression profile of pTRC showed neither a canonical correlation with percent residual tumor (canonical correlation: 0.49, $p = 0.10$) nor with discrete response to PD-1 blockade (canonical correlation: 0.36, $p = 0.60$) (Supplementary Fig. 11E).

While these findings support the notion that raw numbers of tumor-specific TIL are a contributor to therapeutic anti-tumor immunity, clearly transcriptional profiles and consequent functional programs of tumor-specific TIL are just as important. Our prior work suggested that functionally validated MANA-specific TIL from responding tumors showed higher IL7R expression than non-responding tumors[8]. To expand on this observation with the increased number of analyzable cells owing to application of the MANAscore, we next tested for gene expression differences between pTRC in responding and non-responding tumors (Supplementary Data 13; Supplementary Fig. 11F). Indeed, *IL7R* was higher in pTRC of responding relative to non-responding tumors (Fig. 6D). *CXLC13* was also among the top 10 most differentially expressed genes and has previously been reported to correlate with favorable ICB response[10]. Importantly, several genes conventionally associated with effector function (*GZMB*, *GZMA*, *GNLY*, *IFNG*, *LAYN* and *KLRC2*) were upregulated in pTRC from non-responding tumors, with the striking exception of *GZMK*, which was the most upregulated gene in responding tumors. While *GZMB* and *GNLY* have traditionally been associated with effector function, more detailed studies, mostly in the setting of chronic viral infections, have shown a substantial level of overlap in the transcriptional programming between recently-activated effector CD8+ T cells and T cells reaching terminal exhaustion[22,23]. Indeed, the upregulation of effector cytokines along with expression of *LAG3* and *CCL3* by non-responder pTRC is reminiscent of terminally exhausted CD8+ T cells described in the context of chronic antigen stimulation via murine LCMV infection[24]. However, the pTRC from non-responder tumors were enriched in *IFNG* expression whereas *GZMK* and *PDCD1* (PD-1) were notably downregulated, in contrast to classical 'exhausted' CD8+ T cells that generally express high levels of *GZMK* and *PDCD1* and low levels of *IFNG*. These findings are compatible with the notion that pTRC from non-responding tumors exhibit features consistent with an exhausted CD8+ T cell phenotype, but have transcriptional features distinct from conventional exhausted CD8+ T cells in the context of chronic viral infection[25–27]. In support of the antigen-driven nature of the dysfunctional T cell phenotype, the mean checkpoint, cytotoxicity, and TCR signaling scores were all higher in pTRC from non-responding than responding tumors ($p = 0.0069$, 4.0e-7, and 3.0e-13; Fig. 6E–G). Conversely, the upregulation of *IL7R*, *EOMES*, and *CCR7* by responder pTRC more closely resembles a memory precursor or stem-like CD8+ phenotype[25,28,29], although the overall IL7R[lo] nature of pTRC in general suggests an important deficiency in a critical pathway important to maintain robust viral memory populations.

We next assessed the correlation of the ranks of differentially expressed genes between responder and non-responder pTRC in the tumor with that of MANA- vs. flu-specific TIL in three patients with previously validated MANA-specific clones (MD01-004, MD01-005 and MD043-011). Our analysis revealed a strong negative correlation (R = -0.87, $p < 2.2e-16$; Fig. 6H), further supporting the transcriptional similarity of responder tumor-reactive TIL with flu-specific TIL, which are the archetypal long-lived tissue resident memory T cells in the lung. Importantly, this strong correlation shows that that the MANAscore can effectively capture the distinction between responding and non-responding tumors even though training the MANAscore model did not use any responding status information.

The contrasting expression patterns of *GZMK* and *GNLY* in pTRC from responders and non-responders, respectively (Fig. 6D), was of specific interest, since both of these molecules are traditionally associated with CD8+ effector T cell function. To better understand the specific pTRC subsets responsible for the differences observed between responding and non-responding tumors, we performed sub-clustering of all pTRC and identified 9 subclusters (Fig. 7A). *GZMK* was the most differentially-expressed cluster-defining gene for C1 (Fig. 7B), which was also accompanied by expression of other molecules associated with effector function, including *NKG7*, *GZMH* and *CST7*. *TRAT1*, which encodes T cell receptor interacting molecule (TRIM), is also a defining gene of this cluster. TRIM has been shown to regulate TCR expression and stability following antigen-specific stimulation[30]. In addition, *CMC1*, a mitochondrial electron transport chain (ETC) complex IV chaperone protein that acts as a positive regulator of CD8+ T cell activation and differentiation[31] is also upregulated in C1. This subcluster had higher representation among all CD8+ TIL in responder tumors, although this did not reach significance (Fig. 7C; $p = 0.056$). *GNLY*, whose expression was higher in non-responder pTRC (Fig. 6D), was the top cluster-defining gene for C4 (Fig. 7B), however there was no difference in the representation of this subcluster between responder and non-responder pTRC (p = 0.52; Supplementary Fig. 12A). It is worth noting that C4 also exhibited high levels of *KIR2DL4*, *KRT86*, *LAYN*, and *ACP5*, which have all been independently associated with T cell dysfunction[12,32,33], in addition to the effector molecules *GNLY* and *PRF1*, suggesting this may represent effector pTRC approaching terminal dysfunction. The third cluster of interest, C5, closely resembled a stem-like memory population with *IL7R* and *SELL* (CD62L) as the top cluster-

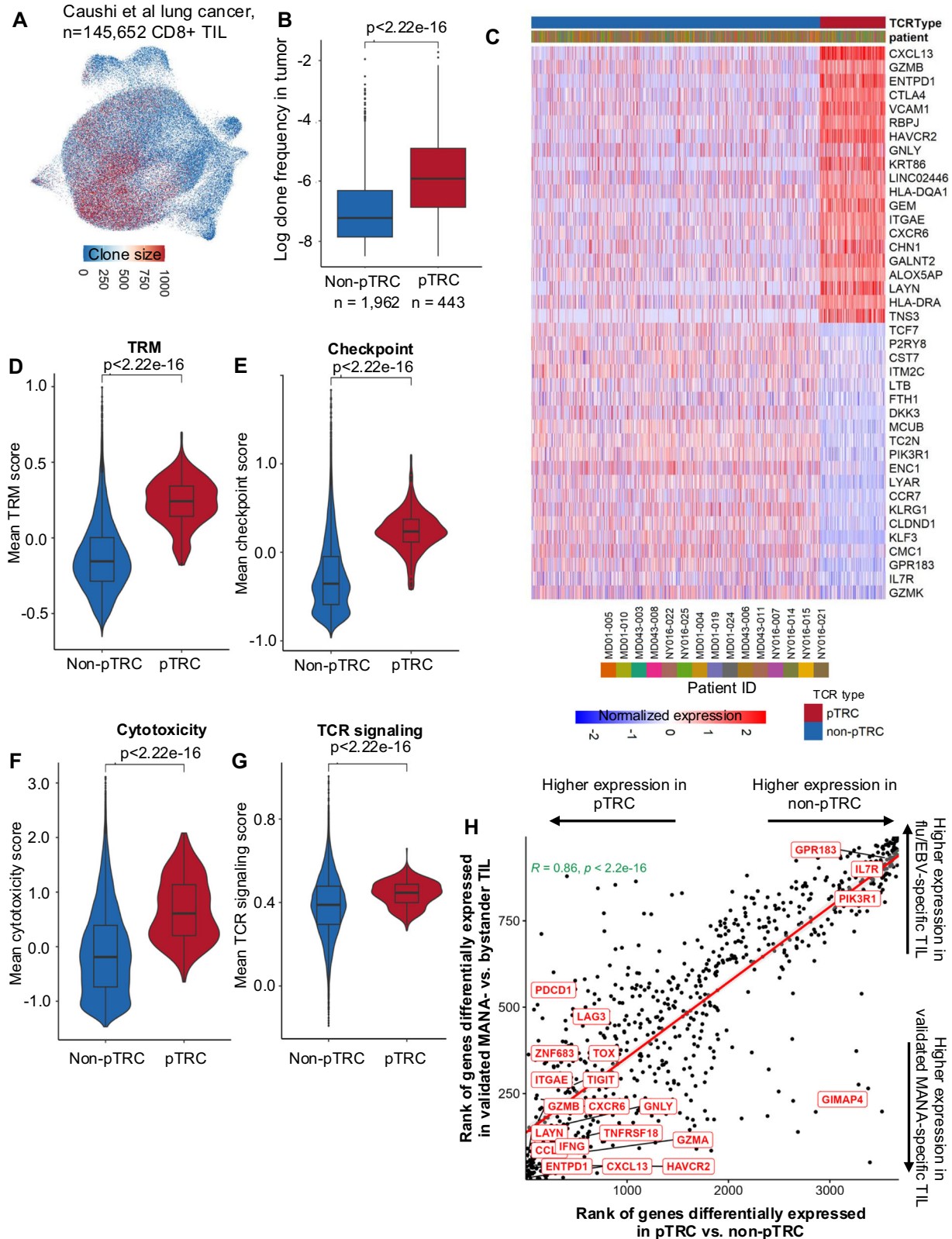

defining genes (Fig. 7B) and this cluster was also enriched in responder pTRC (Fig. 7C). *TCF7* expression was also largely restricted to C5, further supporting the stem-like nature of these cells (Supplementary Fig. 12B). Taken together, these observations suggest that the stem-like subcluster C5 and the effector-like subcluster C1 were the main contributors to the differences observed in pTRC frequency between responder and non-responder tumors described in Fig. 6A, B.

To better understand the relationship between subclusters C1, C4, and C5, we performed trajectory analysis on pTRC from each individual patient (Fig. 7D). We observed a consistent trajectory among all 5 responder tumors originating in C1 (GZMK^hi cluster) and terminating at C5 (stem-like cluster). This is notable owing to the significant upregulation of *IL7R*, *SELL*, *LTB*, and *CCR7* and downregulation of *GZMH*, *GZMK*, *PRF1*, *NKG7*, *GZMB*, *GZMA*, *GNLY*, *EOMES*, and *IFNG* in C5 relative

**Fig. 4 | pTRC reflect known gene programs of tumor-specific TIL. A** UMAP of TIL in NSCLC colored by clonal size, with max clonal size of 1000, the cells with clonal size over 1000 are colored the same with cells with clonal size of 1000. **B** Boxplot showing the proportion of pTRC and non-pTRC in tumor tissue. Data are shown as the average of each clone corresponding to pTRC or non-pTRC (clonal size divided by the total number of TIL in each tumor sample), only the non-pTRC with at least 5 TIL were selected. Comparisons were performed at the clonal level using two-sided Wilcoxon rank-sum test. **C** Top 20 differentially expressed genes in pTRC and non-pTRC based on differential gene expression analysis at the pseudobulk clonal level. **D** Mean TRM signature score, **E** checkpoint score, **F** cytotoxicity score, and **G** TCR

signaling score in pTRC ($n = 443$) and non-pTRC clones ($n = 18,375$) in the tumor. Comparisons were performed at the clonal level using two-sided Wilcoxon rank-sum test. **H** Spearman's correlation between the ranks of differentially expressed genes in pTRC and non-pTRC vs. differentially expressed genes in functionally validated MANA- and bystander (flu- or EBV-specific) TIL. Genes were ranked by log2 fold change from top (upregulated in pTRC or MANA-specific TIL) to the bottom (downregulated in pTRC or MANA-specific TIL). Box plots represent minima maxima as whiskers, the median as the center line within each box, and the interquartile range (IQR) between 25th and 75th percentiles as the bounds of each box.

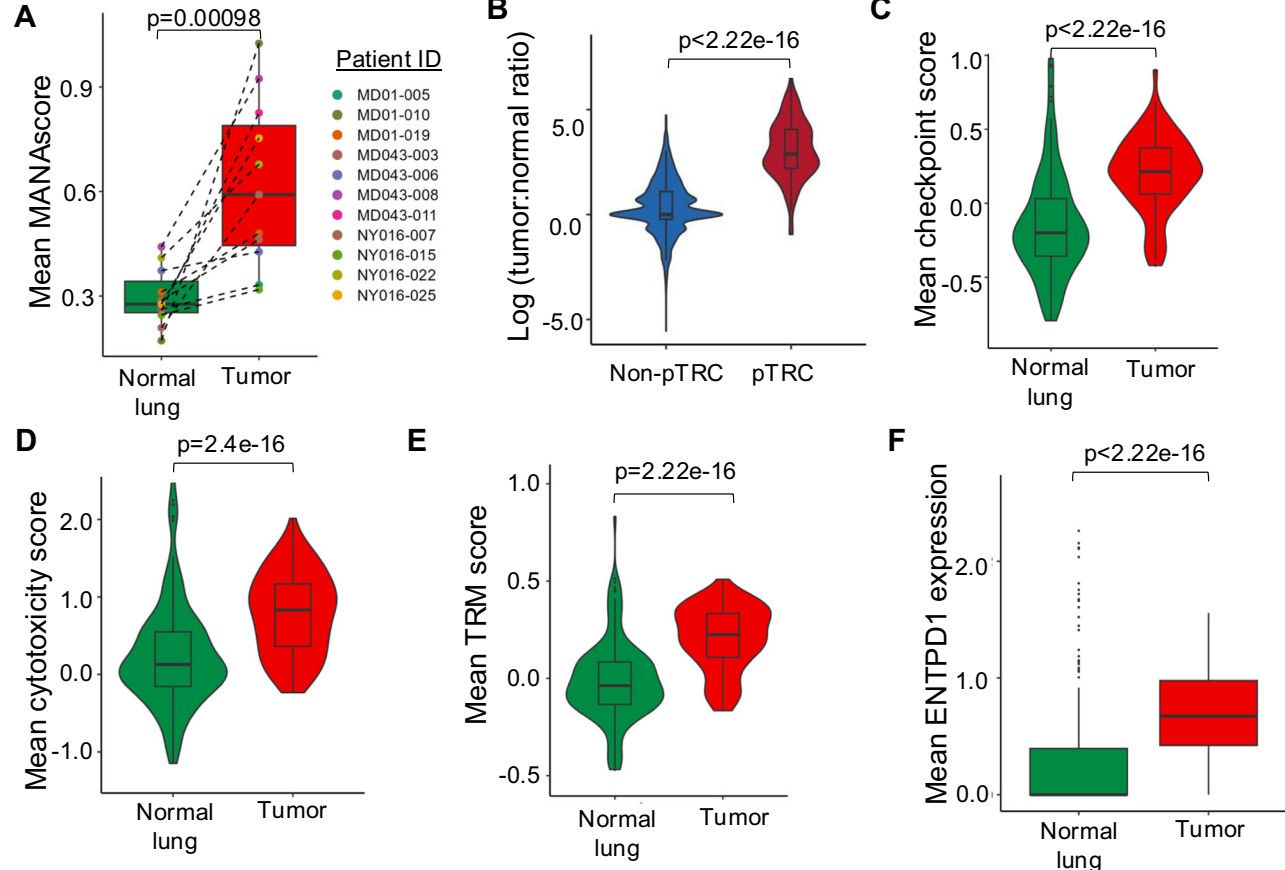

**Fig. 5 | Impact of the TME on pTRC functional programming. A** Boxplot of the average overall MANAscore ($\sqrt{MANAscore\_i^2 + MANAscore\_ni^2}$) in paired adjacent normal lung and tumor ($n = 11$), comparison was performed using two-sided Wilcoxon signed-rank test. **B** Violin plot of the ratio of frequency of tumor: normal for paired pTRC ($n = 156$) and non-pTRC ($n = 2748$) clones. pTRC were identified in the adjacent normal lung by tracking TCR Vβ CDR3 clones corresponding to pTRC in the tumor. Only cells with a TCR Vβ CDR3 clones shared among tumor and adjacent

normal lung were included in this analysis. **C** Average checkpoint, **D** cytotoxicity scores and (**E**) TRM scores of shared pTRC clones ($n = 156$) in normal and tumor tissues. **F** Mean expression of ENTPD1 by pTRC in adjacent normal lung and tumor tissue. Comparisons in B-F were performed with two-sided Wilcoxon rank-sum test. Box plots represent minima maxima as whiskers, the median as the center line within each box, and the interquartile range (IQR) between 25th and 75th percentiles as the bounds of each box.

to C1 (Fig. 7E), suggesting these pTRC begin in an effector-like state and are transitioning to stem-like memory T cells. In 3 of 5 responder tumors, there was also another vector that begins in C1 and ends in C4 (dysfunction cluster), whereas in the other two responder tumors, this vector went through C4 and terminated at C5. Relative to C1, pTRC in C4 had upregulated *GNLY, LAYN, ACP5, KLRC1, TNFRSF18* (*GITR*), *TIGIT, CTLA4,* and *GZMB* (Fig. 7F), suggesting these may be chronically-stimulated effector cells that are approaching terminal dysfunction. These trajectories possibly signify two distinct differentiation paths of GZMK+ pTRC in responder tumors. In contrast, trajectories in the non-responders were notably disordered with no consistent differentiation pathways observed.

## Discussion

Our present study centered on the development and application of a distinct scoring system, MANAscore, which proved highly effective in identifying and characterizing tumor-reactive CD8+ TIL across a spectrum of cancer types and classes of tumor antigen, including neoantigens, cancer-testes antigens, endogenous retroviruses and oncogenic viral antigens, even though our initial goal was to only identify neoantigen specific TIL. These findings suggest common transcriptional programs expressed by tumor-specific T cells that reside in the TME. We previously reported on the distinct transcriptional programming of MANA-specific TIL relative to bystander TIL in NSCLC treated with neoadjuvant PD-1 blockade. Building on this initial

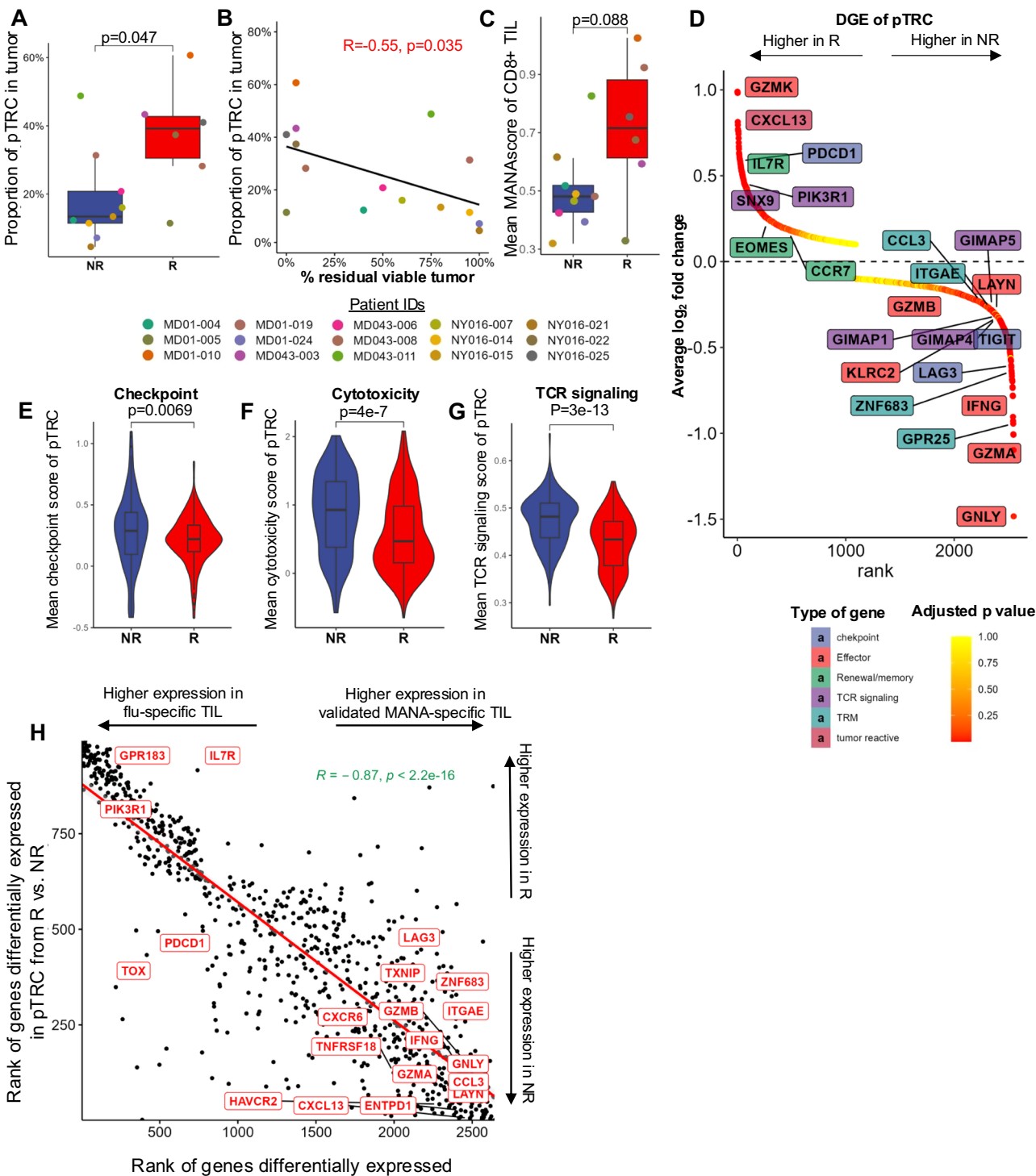

**Fig. 6 | pTRC accurately reflect known functional program differences between validated tumor-reactive TIL from responder and non-responder tumors.**
**A** Proportion of pTRC between patients with responder ($n = 6$) and non-responder ($n = 9$), the comparison was performed using two-sided Wilcoxon rank-sum test. **B** Pearson's Correlation between the proportion of pTRC and residual viable tumor of each patient. **C** Mean MANAscore between responding ($n = 6$) and non-responding tumors ($n = 9$), the comparison was performed using two-sided Wilcoxon rank-sum test. **D** Ranking of differentially expressed genes (Wilcoxon Rank Sum test) between pTRC clones in responder and non-responder patients. **E–G** Mean checkpoint score (**E**), cytotoxicity score (**F**), TCR signaling scores (**G**) in

pTRC in responder ($n = 300$) and non-responder ($n = 143$). Comparisons in (**E–G**) were performed with two-sided Wilcoxon rank-sum test. **H** Spearman's correlation of ranks of gene differentially expression in R vs. NR pTRC and those in validated MANA- vs. flu/EBV-specific TIL, the genes were ranked by the fold change (log2) from top (upregulated in responding pTRC or MANA-specific TIL) to the bottom (downregulated in responding pTRC or MANA-specific TIL). Box plots represent minima maxima as whiskers, the median as the center line within each box, and the interquartile range (IQR) between 25th and 75th percentiles as the bounds of each box.

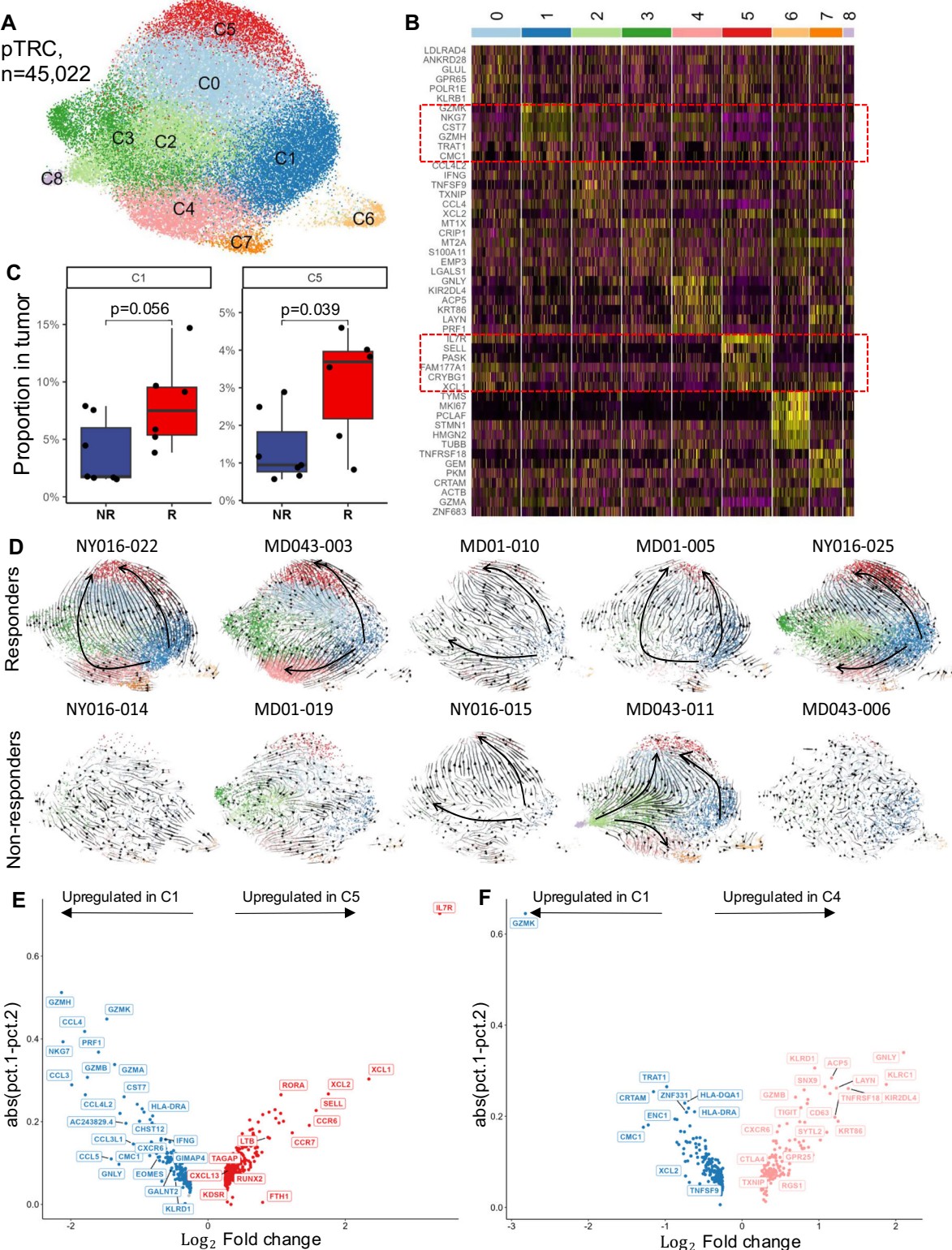

**Fig. 7 | pTRC have distinct trajectories in responder tumors. A** UMAP projection of the expression profiles of 45,022 pTRC from pTRC colored by clusters. **B** Relative expression of top 5 most differentially expressed genes in each cluster. **C** Boxplot of proportion of C1 and C5 pTRC in responding (*n* = 6) and non-responding tumors

(*n* = 9), the comparison was performed using two-sided Wilcoxon rank-sum test. **D** Single patient RNA velocity visualized on the UMAP. **E, F** Differentially expressed genes between C1 and C5 (**E**) as well as C1 and C4 (**F**).

observation, here we developed MANAscore based on expression of only 3 genes, *ENTPD1*, *CXCL13*, and *IL7R*. MANAscore is a useful tool to identify putative MANA-specific TIL without the need for laborious and expensive functional screening of potentially hundreds to thousands of candidate MANA. While this is not the first report of such a score[10,12], MANAscore has the potential to be more accurate than prior gene sets and methods for identifying putative tumor-reactive TIL owing to 1) its superior performance when tested against tetramer- or TCR transfer-validated tumor-specific TIL from resectable cancers relative to other scores, 2) the use of only 3 genes, thus enabling development of multiplex IHC/immunofluorescence and/or simple flow cytometry panels to further study these cells in tumor specimens, and 3) use of combined voting models that improved the previously-published NeoTCR8 score calculated by scGSEA[12].

We first show striking similarities in the gene expression programs of tumor-reactive CD8+ TIL from melanomas and lung cancers, namely that the majority of these cells localize to tissue resident memory clusters and are distinctly clustered away from flu- and EBV-specific TIL. The subsequent development, discovery, and validation of MANAscore highlighted significant heterogeneity both within and between patients that precluded development of a universal threshold for identifying MANAscore^hi cells. Our robust and patient-specific approach allowed us to address these issues of MANAscore heterogeneity to specifically better capture these cells in responding tumors, which tended to have a lower MANAscore in general. In order to further test the robustness of MANAscore, we assessed its ability to identify validated tumor-specific T cells from other published TIL scRNAseq data sets covering multiple other tumor types, including melanoma, head and neck, colorectal and breast cancer. The head and neck TIL were specific for TAA, including the C-T antigen MAGE-A1 and a HERV reported to be expressed at higher levels in some tumors[14].

A distinct class of tumor-specific antigen is represented by oncogenes from integrated human tumor viruses such as HPV and MCPyV. 80% of MCC is caused by integration of a constitutively active fragment of MCPyV T antigen[34]. These MCC on average harbor only 12 total exonic mutations and evidence from tetramer studies supports the notion that the majority, if not the entire anti-tumor T cell repertoire, is indeed directed toward this antigen. We took advantage of this to apply MANAscore to a single-cell RNAseq analysis of MCC TIL whose MCPyV-specific T cells were identified via CITESeq using validated pep/HLA tetramers. Strikingly, roughly half of the MANAscore^hi pTRC were tetramer+ and 99% of the tetramer+ cells were identified by MANAscore. At least some of the MANAscore^hi pTRC that were not tetramer+ were likely specific for T antigen epitopes/HLA alleles not covered by the few tetramers tested by CITEseq. Taken together, MANAscore appears to be remarkably robust in identifying a broad range of tumor antigen classes among many solid cancers with diverse location and biology.

We identified 443 pTRC clones from the lung cancer dataset, which likely represent a substantial reservoir of tumor-specific clones for studying the underlying mechanism of differential ICB response. Specifically, the significant differences in gene expression by pTRC according to treatment response could partially explain the lack of anti-PD-1 responsiveness in some patients despite a high abundance of tumor-reactive TIL. While this was previously addressed by Caushi et al., the higher number of analyzable cells afforded by the MANAscore allowed us to confirm those differences with increased power in addition to providing additional insights on gene expression patterns associated with clinical response to immunotherapy regimens incorporating PD-1 pathway blockers.

Our findings here were able to confirm, with larger numbers of cells, the higher *IL7R* expression by pTRC of responding tumors first reported by Caushi et al. Interestingly, signatures of T cell activation, *IFNG*, and cytolytic machinery were more highly expressed in TIL from non-responding tumors. We hypothesize that this finding reflects a dysfunctional phenotype from the persistence of tumor antigen despite prolonged attempts of the T cells to discharge their machinery to effect tumor killing. A notable exception is *GZMK*, which is the most highly differentially expressed gene in pTRC from responding tumors. Though *GZMK* has been on the list of genes expressed in exhausted cells, it seems to mark a more inflammatory, progenitor-exhausted phenotype rather than a lytic subset of activated CD8+ cells[35]. *GZMK* may also mark an early activation stage of CD8+ TIL with the capacity to expand[36,37]. This is supported by our trajectory analysis showing that pTRC from responding tumors originate in a GZMK^hi state prior to differentiating toward a stem-like program, which is notable owing to prior work that has associated stem-like memory CD8+ T cells with favorable outcome and immunotherapy response[38–40]. Understanding the association of this granzyme to tumor immunity will be an important future endeavor.

Our analysis of pTRC was also benefitted by the availability of high numbers of patient-matched single T cell transcriptomes from adjacent normal lung. Assessment of pTRC in adjacent normal lung suggests that these cells change their functional programming upon entering the TME, thus rendering MANAScore ineffective outside the TME. These findings align with and expand upon existing literature by Yossef et al. showing distinct transcriptional programs of tumor-reactive cells in the periphery relative to those in the tumor[13]

However, our study also identified several caveats that warrant further investigation. While some gene programs identified in pTRC have been described, there were some notable discrepancies in the transcriptional fates observed in pTRC from human tumors with those extensively described in the context of murine LCMV infection. Therefore, the functional nuances and relevance of distinct pTRC subsets remain unclear and necessitate more granular studies of T cell differentiation and functional states specifically in the context of cancer. Additionally, while MANAscore demonstrated notable efficacy in resectable disease contexts, its 54.8% capture rate (17 out of 31 validated clones) in the Lowery et al. dataset hints at the potential for variability in its efficacy in advanced cancers, although the very high capture rate in advanced Merkel cell carcinomas (99.05%) would refute this broad conclusion. While our analysis associated MANAscore with pathologic response following PD-1 blockade in resectable lung cancers, its predictive capabilities in terms of treatment response remain to be fully explored. Thus, prioritization of multi-dimensional analyzes on pre-treatment biopsies, larger validation studies across diverse patient populations, and rigorous assessment of integrated predictive biomarkers will be crucial for advancing our understanding and potential clinical utility of predictive gene signatures. Within a similar framework, understanding the complex interactions of pTRC with all cells in the TME will be crucial for better understanding the underpinnings of response and resistance to ICB. Lastly, in the present study, we identified pTRC as those clones with at least 5 MANAscore^hi TIL, however this absolutely number will heavily depend on the total number of T cells sequenced and will need to be further refined to provide broad recommendations for identifying pTRC across studies.

An additional benefit not explored here is that MANAscore could provide the basis for TCR discovery to facilitate development of engineered T cell therapy. Through comprehensive analyzes, we not only validated previous findings related to tumor-reactive TIL but also revealed potential mechanisms underlying ICB response. The increased precision afforded by MANAscore allowed us to explore the landscape of pTRC in unprecedented detail. In conclusion, our study helps to address the urgent need to better understand and appreciate the phenotype and functional attributes of anti-tumor T cells to facilitate immunotherapy and predictive biomarker development.

## Methods

### Single-cell RNA-seq data collection

Discovery and validation datasets were collected from the Caushi et al. NSCLC cohort (GSE173351)[8] and the Oliveira et al. melanoma cohort (phs001451.v3.p1)[9], which included tumor samples from 15 NSCLC patients and 3 melanoma patients. Detailed patient information is shown in Supplementary Data 1. The Luoma et al. oral cancer (GSE200996)[14] and Lowery et al. metastatic cancer (phs002748.v1.p1)[12] datasets were also used for independent validation. No novel single cell transcriptomic data were generated for this study.

### Single-cell data preprocessing and quality control

Cell Ranger v3.1.0 was used to demultiplex the FASTQ reads, align them to the GRCh38 human transcriptome, and extract their "cell" and "UMI" barcodes. The quality of cells was then assessed based on (1) the number of genes detected per cell and (2) the proportion of mitochondrial gene/ribosomal gene counts. Low-quality cells were filtered if the number of detected genes was below 250 or more than 3 standard deviations from the median gene number of all cells. Cells were filtered out if the proportion of mitochondrial gene counts was higher than 10% or the proportion of ribosomal genes was less than 10%. For single-cell VDJ sequencing, only cells with full-length sequences were retained. Dissociation/stress associated genes[41,42], mitochondrial genes (annotated with the prefix "MT-"), high abundance lincRNA genes, genes linked with poorly supported transcriptional models (annotated with the prefix "RP-")[43] and TCR (TR) genes (TRA/TRB/TRD/TRG, to avoid clonotype bias) were removed from further analysis. In addition, genes that were expressed in less than five cells were excluded.

### Single-cell data integration and clustering

Only CD8+ T cells were included in this study. To select for CD8+ T cells, SAVER[44] was used to impute dropouts by borrowing information across similar genes and cells. A density curve was fitted to the log2-transformed SAVER-imputed *CD8A* expression values (using 'density' function in R) of all cells from all samples. A cutoff was determined as the trough of the bimodal density curve (i.e., the first location where the first derivative is zero and the second derivative is positive). All cells with log2-transformed SAVER imputed *CD8A* expression larger than the cutoff were defined as CD8+ T cells. Seurat (3.1.5)[45] was used to normalize the raw count data, identify highly variable features, scale features, and integrate samples. Principal component analysis (PCA) was performed based on the 3000 most variable features identified using the vst method implemented in Seurat. Gene features associated with type I Interferon (IFN) response, immunoglobulin genes and specific mitochondrial related genes were excluded from clustering to avoid cell subsets driven by the above genes[43]. The 'FindIntegrationAnchor' function was used for selecting anchors, which were then utilized in the integration process via the 'IntegrateData' function. Dimension reduction was done using the RunUMAP function. Cell markers were identified using a Wilcoxon rank sum test. Genes with adjusted *p*-value < 0.05 were retained. Clusters were labeled based on the expression of the top differential gene in each cluster as well as canonical immune cell markers. For the metastatic cancer (phs002748.v1.p1)[12] dataset, the samples from patient 4421 were removed due to strong batch effect generated by the samples from this patient.

### Computational development and validation of MANAscore

All three patients from the Oliveira et al. melanoma cohort with both validated MANA- and EBV-/InfluenzaA-specific T cells were used for training. Ground truth MANA-specific T cells were labeled as '1' and EBV-/InfluenzaA-specific T cells as '0'. We used two types of input data: quantile normalized SAVER imputed expression matrix and Seurat normalized gene expression matrix without SAVER imputation. These data sets were utilized to train predictive models using two algorithms: random forest and linear regression. For random forest algorithm, we performed GridSearchCV to choose the best parameters (max_features: 3 and n_estimators: 100) based on the initial parameter combinations (max_features: 1, 2, 3, and n_estimators: 100, 500, 1000, 2000, 2500). 12 single-patient models were initially constructed (resulting from the combination of 3 training patients, 2 data types and 2 algorithms). To enhance the models' robustness, an ensemble method, 'voting', was used to combine the performances of 6 models based on imputed data and non-imputed data respectively. 'VotingClassifier' in 'sklearn.enemble' was used to build voting models with soft voting. For model construction, 80% of cells of the ground truth labels was used for training, and the remaining 20% data was used for later testing models.

Melanoma associated antigen-specific T cells (MAA) shared common features with MANA-specific T cells, so all melanoma associated antigen-specific T cells and 20% of the EBV-/InfluenzaA-specific T cells were also used as a test dataset. The models trained using melanoma data were subsequently applied to NSCLC test data[8] and also independently validated on oral cancer[14], metastatic cancer[12] and MCPyV⁺ merkel cell carcinoma datasets. In order to assess the performance of MANAscore in comparison to previously published models, including NeoTCR8[12] and the single gene model (CXCL13)[10], as well as two other single gene models (*ENTPD1* and *IL7R*), we generated scores with respective signatures. The pROC package was utilized to draw the receiver operating curves (ROCs) and to evaluate the areas under the curves (AUCs). We defined putative tumor-reactive cells (pTRC) as CD8+ TIL in the clones which are with at least 5 MANAscore^hi cells. TIL with clones shared among different patients were excluded from pTRC. To simplify visualization and comparison of MANAscore across different biological compartments, we computed the overall MANAscore combining both MANAscore_i and MANAscore_ni into one score ($\sqrt{\text{MANAscore\_i}^2 + \text{MANAscore\_ni}^2}$).

### Pseudobulk gene expression analysis

PCA was performed on a standardized pseudobulk gene expression profiles, where each feature was standardized to have a mean of zero and unit variance. Combat function in the "sva" R package[46,47] was applied to address potential batch effects on the normalized pseudobulk profile. Highly variable genes (HVGs) were selected for each cell cluster by fitting a locally weighted scatterplot smoothing (LOESS) regression of standard deviation against the mean for each gene and identifying genes with positive residuals. For each sample, all cell clusters were then concatenated by retaining each cluster's HVGs to construct a concatenated gene expression vector consisting of all highly variable features identified from different cell clusters. Each element in this vector represents the pseudobulk expression of a HVG in a cell cluster. Samples were embedded into the PCA space based on these concatenated gene expression vectors. Canonical correlation[48,49] between the first two PCs (i.e., PC1 and PC2) and a covariate of interest (i.e., response status) was calculated. Permutation test was used to assess the significance by randomly permuting the sample labels 10,000 times.

### Differential analyzes comparing pTRC vs. non-pTRC and pTRC in responders vs. non-responders

For each clone, all cells were first concatenated to construct a concatenated gene expression matrix, then clone level Seurat object was generated, followed by normalization. Differential-expression tests for comparison of pTRC vs. non-pTRC and pTRC in responders vs. non-responders were performed using FindAllMarkers functions in Seurat with Wilcoxon rank-sum test on pseudobulk clone expression value.

## Signature score generation

The signatures including checkpoint, cytotoxicity, TRM signature, stemless, TCR signaling related genes were collected from published paper (Supplementary Data 12). The scores were computed using AddModuleScore function in Seurat.

## Statistical analysis

R v.4.0.2 and Python 3.6 were used for statistical analyzes. No statistical method was used to predetermine sample size. Patients were not randomized. Neither investigators nor patients were blinded to treatments in the discovery and validation cohorts. Numerical data are reported as mean ± s.d. For analysis requiring two-sided nonparametric and parametric calculations, Mann–Whitney $U$-test and Student's $t$-test, respectively, were used for nonpaired observations. Wilcoxon signed-ranked test was used for nonparametric calculations comparing matched samples. Pearson´s $r$ was used to correlate raw data values of the indicated variables depicting linear relationships. One-way analysis of variance was used for multiple comparisons, and $P$ values were adjusted using Benjamini-Hochberg (BH) method for multiple comparisons where appropriate. A $P$ value threshold of 0.05 was considered statistically significant.

## Reporting summary

Further information on research design is available in the Nature Portfolio Reporting Summary linked to this article.

## Data availability

All processed and de-identified single-cell data from the neoadjuvant-treated human samples are available in the Gene Expression Omnibus at the link https://www.ncbi.nlm.nih.gov/geo/query/acc.cgi?acc= GSE176022. In addition to the processed single-cell data, the raw scRNA-seq–TCR-seq data are available in the European Genome-phenome Archive at https://ega-archive.org/datasets/ EGAD00001007728. Owing to the personal, sensitive and inherently identifying nature of raw genomic data, access to raw RNA-seq–TCR-seq data is controlled and may require institutional data use agreements. Full instructions to apply for data access can be found at https://ega-archive.org/access/data-access. All data needed to evaluate the conclusions in the manuscript are present in the paper or the supplementary materials.

## Code availability

Scripts and a package to reproduce the analyzes performed in this study can be found at https://github.com/BKI-immuno-KNS/ MANAscore[50].

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

## Acknowledgements

The Mark Foundation for Cancer Research (DMP), Bloomberg~Kimmel Institute for Cancer Immunotherapy (DMP, SLT, and KNS), The Mark Foundation Center for Advanced Genomics and Imaging (DMP), Cancer Research Institute (KNS), Lung Cancer Foundation of America (KNS), LUNGevity (KNS), American Lung Association (KNS), Swim Across America (KNS), Commonwealth Foundation (KNS), Bristol-Myers Squibb (DMP, PMF), National Institutes of Health grants R37CA251447 (K.N.S.), R01HG010889 (H.J.), R01HG009518 (H.J.), R01HG013409 (H.J.), P01 CA225517 (PHN), T32 CA080416 (SJ), and P30 CA006973 (KNS, DMP, SLT), Kelsey Dickson Team Science Courage Research Award: Advancing New Therapies for Merkel Cell Carcinoma (MCC) (PHN), MCC Patient Gift Fund (PHN), National Foundation for Cancer Research (SLT and PHN).

## Author contributions

K.N.S., H.J., D.M.P, S.L.T., and P.N.. conceptualized the study. H.J. supervised the development and implementation of computational methods. Z.Z., T.Z., J.Z., S.L., S.C., B.Z., Y.Z., J.W, D.S., R.K., C.D.C., T.H.P., and S.J. carried out the investigation. Z.Z., T.Z., J.Z., S.L., B.Z., Y.Z., and J.W. carried out the data analysis. K.N.S. and Z.Z. wrote the original manuscript draft. All authors reviewed and edited the manuscript draft. P.M.F., K.N.S., H.J., S.L.T and C.D.C. acquired the funding.

## Competing interests

P.M.F. receives research support from AstraZeneca, BioNtech, Bristol-Myers Squibb, Novartis, Regeneron, and has been a consultant for AstraZeneca, Amgen, Bristol-Myers Squibb, Iteos, Novartis, Star, Surface, Genentech, G1, Sanofi, Daiichi, Regeneron, Tavotek, VBL Therapeutics, Sankyo, and Janssen and serves on a data safety and monitoring board for Polaris. S.Y. receives research funding from Bristol-Myers Squibb/Celgene, Janssen, and Cepheid, has served as a consultant for Cepheid, and owns founders' equity in Brahm Astra Therapeutics and Digital Harmonic. K.N.S. and D.M.P. have filed for patent protection on the MANAFEST technology (serial No. 16/341,862). D.M.P. is a consultant for Compugen, Shattuck Labs, WindMIL, Tempest, Immunai, Bristol-Myers Squibb, Amgen, Janssen, Astellas, Rockspring Capital, Immunomic, Dracen and owns founders' equity in Clasp Therapeutics, WindMIL, Trex, Jounce, Enara, Tizona, Tieza, RAPT and receives research funding from Compugen, Bristol-Myers Squibb, and Enara. K.N.S. has received travel support/honoraria from Illumina, Inc., receives research funding from Bristol-Myers Squibb, Abbvie, and Astra Zeneca, and owns founder's equity in Clasp Therapeutics. S.L.T receives consulting fees from Bristol Myers Squibb, Dragonfly Therapeutics, PathAI, and Regeneron; receives research grants from Bristol Myers Squibb; has stock options in Dragonfly Therapeutics; and has a patent related to the treatment of MSI-high cancers with anti-PD-1.The terms of all these arrangements are being managed by Johns Hopkins University in accordance with its conflict-of-interest policies. All other authors declare that they have no competing interests.
