## [Transparent Peer Review file · Nature Communications]

A minimal gene set characterizes TIL specific for diverse tumor antigens across different cancer types

Corresponding Author: Dr Kellie Smith

Version 0:

Reviewer comments:

Reviewer #1

(Remarks to the Author)

In this manuscript, Zheng et al provide an integrated reanalysis and extension of their prior work identifying and annotating mutation associated neoantigen T cells by combining two datasets in which MANAFEST was performed for melanoma and NSCLC patients. The authors provide a novel signature that could help identify MANA- T cells, and provide detailed in silico analyses evaluating the performance of their three gene signature, more detailed phenotyping of the T cells identified, and more correlation of MANA- T cells with response to immunotherapy. Overall, I think this manuscript significantly advances our understanding of tumor reactive T cells. While the proposed markers (IL7R, CD39, and CXCL13) have been identified by many groups, how best to use all three markers to identify tumor specific T cells has not previously been established. Limitations include the limited sample size and a limited connection between their minimal gene set and clinical outcomes. Despite these minor concerns, I feel that the manuscript is largely ready for publication in its current state.

I initially had many technical questions about the robustness of the computational approaches of the figures, but after review of the code provided by the authors provided on github, I have only minor concerns. In Figure 1, the authors could better describe how anchoring was used for integrated analysis in the methods. In Figure 2, it would be helpful to indicate in the legend that the green line in 2E represents the linear correlation. In Figure 3, insufficient rationale and explanation is provided as to when the authors begin to calculate the overall Manascore from both imputed and non-imputed inputs. In Figure 5, given the heterogeneity observed across patients when assessed with single cell RNA sequencing, I'm not sure whether analysis of two patients provides clinically meaningful insights.

(Remarks on code availability)

Reviewer #2

(Remarks to the Author)

Zheng, Z. and colleagues present a very intriguing study investigating the transcriptional program(s) of mutation-associated, neoantigen-reactive T cells. In this work they utilize an interesting collection of ground truth datasets with paired scRNA and scTCR sequencing to identify a signature "specific" to neoantigen reactive T cells in lung and skin cancers. The authors do diligent immunological studies to further characterize specifically what kinds of T cells this score identifies, which appears to be putative tumor reactive clones. The various datasets and associated analyses in this study are quite convincing and I'm left somewhat surprised that only 3 genes are necessary to support such robust results. This work does get a little confusing as to whether it's focused on neoantigen reactive T cells or just tumor reactive clones. This makes it difficult to see if the neoantigen specificity can validate outside of the two additional studies (not training set) that were included. There are a few major and minor comments below, but overall this is a well done study and the authors were quite thorough in finding convincing support for their conclusions.

Major Comments:

Although imputation can be "accurate", it is not part of a standard workflow and it should be done with some caution,

because not all genes are zero inflated. There will be some overcompensation, since SAVER does not make any assumptions about how much zero inflation exists, just that there are zeros that should be non-zero. By performing both integration and imputation, there are multiple manipulations to the raw data, which should be justified somewhere. Can you provide some evaluation of what and how much was added after imputation? How much of your remaining results will depend on these artificially amplified signals? Additionally, the methods should specify the method of integration, as there are multiple from which to choose.

It's not very clear, but are only these three genes included in the ensemble ML methods? Random forest is suboptimal in this situation, since it is primarily used to test different combinations of features for decision trees. Given this, it's not an entirely fair comparison between a 3 gene set with RF and a large gene set like NeoTCR8. This obviously doesn't change the AUC, but I would be curious how 243 gene NeoTCR8 genes would perform using the described ensemble methodology, rather than the ROC method described in their methodology.

Figure 3, while very interesting, doesn't support this being a MANA score, since viral oncogenes are not mutations nor are they neoantigens. Yes, the MANA score is convincingly specific to the MCPyV antigen, but this diminishes the significance of the score being specific to mutation based neoantigens.

Many of the significance tests between group frequencies (ie Fig. 5G) utilize a Wilcoxon signed-rank test, but the cells are not paired measurements, so the test should be Wilcoxon rank-sum test. Figure 5A correctly uses the signed rank test, but proper tests should be utilized, especially paired vs. unpaired tests.

In the end, although the MANAscore is derived from MANA T cells, it clearly identifies more than just MANA T cells, which is fine. However, it would be less confusing for the presented story to consider a different naming convention, or make it more clear from the beginning (not just the title), that the MANA score helps identify pTRCs. This transition from MANA T cell identification to just pTRC identification was not easy to catch until around figure 4. By the time the reader makes it to figure 5, MANA is just the score, not the cell of interest anymore.

Minor Comments:

Figure 1D legend, I believe you meant to put TAA instead of MAA for the greens. In the text you call it Melanoma Associated Antigen (MAA) and that's what the legend has, but the figure text says TAA?

Figure 4H should use spearman rho rather than pearson correlation, as ranks are not continuous values and spearman already utilizes rank. Of course, this will make the p-value even more significant, but it is technically the better metric in this scenario.

(Remarks on code availability)

The github page appears to be more of a catalog of analyses performed. For an experienced programmer, there is enough information contained on this page to reproduce most of the manuscript. The code is not necessarily formatted to "run" like a functional package. Although this is fine to me, it does make for a little extra effort if someone wanted to run the MANAscore, considering it's not calculated from just those three genes. In that case, some documentation is likely needed on dependencies and how to run the model. There is no documentation on installing and running in the README file.

Version 1:

Reviewer comments:

Reviewer #2

(Remarks to the Author)

I have reviewed the authors' comments to reviewers and believe they have sufficiently addressed the concerns from myself and reviewer #1. I thank the authors for performing the additional experiments requested and am intrigued by the implications of these results. I have no further comments/concerns that need addressing.

(Remarks on code availability)

I have reviewed the code on Github. I have the same comments as before. The code posted to github is not formatted to run like a function, but more so formatted as a type of "notebook" of how the analyses were performed. While I don't think it is necessary to make this into a package, I would recommend the authors to consider this option. 1) Your code is not licensed on Github, so there is no legal obligation of citing your work, if your code is used. 2) Your work is easier to benchmark in future studies if there is a functional package attached. That way future studies also perform the MANAscore analysis the way you intended. This is just my suggestions, though.

RESPONSE TO REVIEWERS

We would like to thank the reviewers and editors for their thorough analysis of our study, and for considering our study for publication in *Nature Communications*. We are especially heartened by the reviewers' appreciation of our "diligent immunological studies" and "convincing" analyses. Below, we have put together a thorough point-by-point response to the issues raised and have modified the manuscript accordingly, resulting in a much-improved study.

Reviewer #1:

1. I initially had many technical questions about the robustness of the computational approaches of the figures, but after review of the code provided by the authors provided on github, I have only minor concerns.

We thank the reviewer for this comment. It is our goal to be completely transparent in our methods to ensure data robustness and reproducibility. We provided even more clarifications to the code to ensure smooth reproducibility.

2. In Figure 1, the authors could better describe how anchoring was used for integrated analysis in the methods.

For data integration in Figure 1, we added the anchoring details in the methods (page 19, lines 518-519).

3. In Figure 2, it would be helpful to indicate in the legend that the green line in 2E represents the linear correlation.

We added the meaning of the green line in Fig. 2E to the legend (page 28, lines 788-789).

4. In Figure 3, insufficient rationale and explanation is provided as to when the authors begin to calculate the overall Manascore from both imputed and non-imputed inputs.

Thank you for pointing this out. The rationale for calculating the overall MANAscore from both imputed and non-imputed MANAscore is explained below.

During our score development, we generated two scores (MANAscore_i and MANAscore_{ni}) based on applying our ensemble voting approach to SAVER-imputed and non-imputed data respectively. These two scores are highly correlated with each other, ranging from R=0.68 in MD01-019 to R=0.9 in MD01-004 and NY016-021 (as shown in Fig. S3), and they were used to jointly define the MANAscore^{hi} cells as in Fig. 2E. However, for visualizing cells' MANAscores on UMAP or comparing MANAscore across different biological compartments (e.g. tumor vs normal lung), it will be easier to run analysis and present the results using a single score rather than two different scores. Therefore, for visualization and compartment comparisons, we developed an overall MANAscore ($\sqrt{\text{MANAscore}_i^2 + \text{MANAscore}_{ni}^2}$) which combines MANAscore_i and MANAscore_{ni} into one score.

In our five test and validation datasets, we observed that neither of MANAscore_i and MANAscore_{ni} was consistently superior across all datasets, but the combined overall MANAscore robustly showed performance closer to the better one of MANAscore_i and MANAscore_{ni} in most datasets. Specifically, in the melanoma and NSCLC datasets where true positive and true negative information are available for generating ROC curves, we can notice that MANAscore_i outperforms MANAscore_{ni}, and AUC of overall MANAscore

$(\sqrt{\text{MANAscore}_i^2 + \text{MANAscore}_{ni}^2})$ is between MANAscore_{ni} and MANAscore_i , but consistently closer to MANAscore_i except MD01-005 in NSCLC validation data (Fig. R1).

Figure R1. Comparison of imputed and non-imputed MANAscore on test and validation datasets.

In those three other datasets including Luoma et al. oral cancer, Lowery et al. metastatic cancers, and the novel MCPyV-positive Merkel cell carcinoma data included in our present study, we did not have true negative information for generating ROC curves, therefore we compared different scores by calculating the rankings of MANAscore_i , MANAscore_{ni} and overall MANAscore of validated tumor reactive TIL for each individual patient (Fig. R2). Smaller ranking would indicate better performance. We noticed that MANAscore_i was better than MANAscore_{ni} in the Merkel cell carcinoma data, while the opposite is true in the oral cancer and multiple-type metastatic cancer data set. Therefore, neither of these two scores was consistently superior among all datasets. Importantly, however, the combined overall MANAscore was always closer to the better one of MANAscore_i and MANAscore_{ni} , demonstrating its robust performance. Because of this, we used the combined overall MANAscore ($\sqrt{\text{MANAscore}_i^2 + \text{MANAscore}_{ni}^2}$) for the downstream visualization and comparisons instead of using two individual scores. To clarify this decision, we have added Fig. S9A and a better description of this method to page 7, lines 191-192, page 10, lines 273-275 and page 11, lines 276-278.

Figure R2. Ranking of tumor-reactive CD8+ TIL using different MANAscore models.

5. In Figure 5, given the heterogeneity observed across patients when assessed with single cell RNA sequencing, I'm not sure whether analysis of two patients provides clinically meaningful insights.

We thank the reviewer for bringing up this important point. We originally chose to show this because, while they are the only patients for whom we had matched lymph node specimens for single cell analysis, these patients coincidentally represented completely opposing ends of the response spectrum: MD01-005 with 0% residual viable tumor at the time of surgery and MD043-011 with 75% residual viable tumor at the time of surgery. But after reflecting on the reviewer's point, we agree that this does not add much to the story and is limited by anecdotal evidence in only 2 patients, so we have removed the two figure panels containing the lymph node data (original Figures 5F and 5G).

Reviewer #2:

1. This work does get a little confusing as to whether it's focused on neoantigen reactive T cells or just tumor reactive clones. This makes it difficult to see if the neoantigen specificity can validate outside of the two additional studies (not training set) that were included.

Thanks for pointing this out. Our MANAscore was initially developed and trained from MANA-specific TIL with the goal of identifying only MANA-specific TIL, so we call our model 'MANAscore'. However, as the reviewer points out, much of the available "ground truth" data are on TIL recognizing other classes of tumor antigens, thus allowing us to explore the performance of MANAscore on other tumor antigen types. As presented in our study, our MANAscore indeed has high accuracy to identify CD8+ TIL that recognize several different classes of tumor antigens, not just neoantigens. We have clarified this point in lines 35-36, 88-90, and 392. In fact, we believe this finding supports the notion that the microenvironment context may be more important in defining transcriptional programs than of the tumor antigen class for which the T cell is recognizing.

2. Although imputation can be "accurate", it is not part of a standard workflow and it should be done with some caution, because not all genes are zero inflated. There will be some overcompensation, since SAVER does not make any assumptions about how much zero inflation exists, just that there are zeros that should be non-zero. By performing both integration and imputation, there are multiple manipulations to the raw data, which should

be justified somewhere. Can you provide some evaluation of what and how much was added after imputation? How much of your remaining results will depend on these artificially amplified signals? Additionally, the methods should specify the method of integration, as there are multiple from which to choose.

We thank the reviewer for this excellent point, and we have partially addressed this concern above in the reply to Reviewer 1, #4, as it relates to model construction. Moreover, before we conducted integration, we performed imputation to select cells with CD8A expression based on the bimodal distribution of imputed CD8A, but we did not use the imputation matrix for the integration. To clarify this, we have modified and added additional methods of integration to the manuscript, page 19, lines 508-513 and 518-519.

3. It's not very clear, but are only these three genes included in the ensemble ML methods? Random forest is suboptimal in this situation, since it is primarily used to test different combinations of features for decision trees. Given this, it's not an entirely fair comparison between a 3 gene set with RF and a large gene set like NeoTCR8. This obviously doesn't change the AUC, but I would be curious how 243 gene NeoTCR8 genes would perform using the described ensemble methodology, rather than the ROC method described in their methodology.

We thank the reviewer for bringing up this point. First, we would like to clarify that, as a non-parametric approach capable of modeling complex non-linear functions, random forest not only explores different combinations of features, but can also explore different ways to partition the sample space using a given feature set. Even with only three features (i.e., 3 genes), there can be many different ways to partition the sample space (i.e., partition cells) for modeling the non-linear function, and the random forest approach therefore is suitable for our situation. Next, to see how 243-gene NeoTCR8 genes would perform using our ensemble approach, we applied our ensemble methodology (combined voting model) to the 243 genes in NeoTCR8 (we will call this 'MANAscore-NeoTCR8') and compared its performance to our 3-gene MANAscore, as well as to the original published NeoTCR8 using scGSEA. In our test dataset, the MANAscore_NeoTCR8 models actually have modestly better performance than our 3-gene MANAscore models, while they perform worse than MANAscore in the validation data (Fig. R3). We then found that the MANAscore-NeoTCR8 model significantly outperforms the original NeoTCR8 as published by Lowery et al (i.e. using scGSEA) in both training and validation data.

Figure R3. Performance of different models and gene sets on test (top) and validation (bottom) datasets.

We then calculated the rankings of tumor reactive CD8+ TIL in three additional datasets that did not have non-tumor-specific TIL identified (thus ROC curves cannot be generated) for different scoring models, including the original NeoTCR8 score calculated by scGSEA. We calculated the rankings by patient, with a smaller ranking indicating better performance (Fig. R4). We notice that MANAScore performs better than the MANAScore_NeoTCR8 in all 3 datasets. Interestingly, NeoTCR8 calculated by scGSEA only performs best in the metastatic cancer dataset, i.e. the dataset from which it was first developed, thus opening the possibility that this may be due to overfitting. Moreover, there is no question that a 3-gene MANAScore is significantly more parsimonious than the 243-gene NeoTCR8, in many regards.

Figure R4. Ranking of tumor-reactive CD8+ TIL using different models.

Taken together, MANAScore performs better than MANAScore-NeoTCR8 in 4 of 5 datasets (melanoma test data is the exception, from which 80% of the training data came) and MANAScore-NeoTCR8 outperforms NeoTCR8 in 4 of 5 datasets (the exception is the dataset from which it was first developed). We feel these findings support use of MANAScore over NeoTCR8.

We have added discussion points to address these findings as Fig. S2D and S9, and have provided a brief description about these caveats and comparisons to page 7, line 172-175, and page 15, line 403-404.

4. Figure 3, while very interesting, doesn't support this being a MANA score, since viral oncogenes are not mutations nor are they neoantigens. Yes, the MANA score is convincingly specific to the MCPyV antigen, but this diminishes the significance of the score being specific to mutation based neoantigens.

Thanks for your comments. Please see our reply to #1 above. We have reframed our study to emphasize this fact: our gene signature can identify tumor-reactive TIL spanning different categories of tumor antigen, not just those recognizing mutation-associated neoantigens. Please see modifications/discussion of this point in lines 35-36, 88-90, and 392.

5. Many of the significance tests between group frequencies (ie Fig. 5G) utilize a Wilcoxon signed-rank test, but the cells are not paired measurements, so the test should be Wilcoxon rank-sum test. Figure 5A correctly uses the signed rank test, but proper tests should be utilized, especially paired vs. unpaired tests.

Per Reviewer 1, comment #5, we have removed the lymph node analyses (Figs. 5F and 5G). We have additionally clarified the test method in the legend of Figure 5, page 31, lines 825-826 and line 831.

6. In the end, although the MANAScore is derived from MANA T cells, it clearly identifies more than just MANA T cells, which is fine. However, it would be less confusing for the presented story to consider a different naming convention, or make it more clear from the beginning (not just the title), that the MANA score helps identify pTRCs. This transition from MANA T cell identification to just pTRC identification was not easy to catch until around figure 4. By the time the reader makes it to figure 5, MANA is just the score, not the cell of interest anymore.

This is a great point. We agree that the transition from MANA-specific TIL identification to a broader application of the MANAScore for identifying pTRC could be more clearly communicated. We have therefore better articulated this point in lines 35-36, 88-90, and 392.

Minor Comments:

7. Figure 1D legend, I believe you meant to put TAA instead of MAA for the greens. In the text you call it Melanoma Associated Antigen (MAA) and that's what the legend has, but the figure text says TAA?

We changed the figure to MAA.

8. Figure 4H should use spearman rho rather than pearson correlation, as ranks are not continuous values and spearman already utilizes rank. Of course, this will make the p-value even more significant, but it is technically the better metric in this scenario.

We've amended the correlation analysis using Spearman's correlation in Fig. 4H and Fig. 6H.

Remarks on code availability:

9. The github page appears to be more of a catalog of analyses performed. For an experienced programmer, there is enough information contained on this page to reproduce most of the manuscript. The code is not necessarily formatted to "run" like a functional package. Although this is fine to me, it does make for a little extra effort if someone wanted to run the MANAScore, considering it's not calculated from just those three genes. In that case, some documentation is likely needed on dependencies and how to run the model. There is no documentation on installing and running in the README file.

This is an excellent point. We've uploaded the imputed and non-imputed gene matrix in patients with ground truth, as well as added the necessary description of running MANAScore to the README file.

RESPONSE TO REVIEWERS

We would like to thank the reviewers and editors for their thorough analysis of our study, and for considering our study for publication in *Nature Communications*.

Reviewer #2:

I have reviewed the authors' comments to reviewers and believe they have sufficiently addressed the concerns from myself and reviewer #1. I thank the authors for performing the additional experiments requested and am intrigued by the implications of these results. I have no further comments/concerns that need addressing.

We thank the reviewer's comment, and we're pleased to have addressed the concerns raised.

I have reviewed the code on Github. I have the same comments as before. The code posted to github is not formatted to run like a function, but more so formatted as a type of "notebook" of how the analyses were performed. While I don't think it is necessary to make this into a package, I would recommend the authors to consider this option. 1) Your code is not licensed on Github, so there is no legal obligation of citing your work, if your code is used. 2) Your work is easier to benchmark in future studies if there is a functional package attached. That way future studies also perform the MANAscore analysis the way you intended. This is just my suggestions, though.

We added MIT license, and reorganized the Github repository, and we put the files related to model construction under a separate folder named 'MANAscore', there includes:

1. Script 'MANAscore.py' for model construction.
2. Folder '3gene' which are three-gene imputed and non-imputed matrix of ground truth for training and test data in melanoma, validation data in lung cancer.
3. Folder 'models' including imputation combine voting model (voting_i_classifier.pkl.gz) and non-imputation combine voting model (voting_ni_classifier.pkl.gz) saved for MANAscore prediction.

And we put all R scripts for analysis including data integration, data preprocessing for MANAscore prediction, model evaluation and differential gene/signature analyses under folder named 'Analyses'.

The package MANAscore is now available for people who are interested in our method to easily install.

We also include a tutorial (<https://bki-immuno-kns.github.io/MANAscore/MANAscore-Tutorial.html>) for users who are interested in our MANAscore.